# EFFECT OF MODEL AND PRETRAINING SCALE ON CATASTROPHIC FORGETTING IN NEURAL NETWORKS

**Vinay Ramasesh, Aitor Lewkowycz, and Ethan Dyer**
Google Research, Blueshift
{ramasesh,alewkowycz,edyer}@google.com

## ABSTRACT

Catastrophic forgetting presents a challenge in developing deep learning models capable of continual learning, i.e. learning tasks sequentially. Recently, both computer vision and natural-language processing have witnessed great progress through the use of large-scale pretrained models. In this work, we present an empirical study of catastrophic forgetting in this pretraining paradigm. Our experiments indicate that large, pretrained ResNets and Transformers are significantly more resistant to forgetting than randomly-initialized, trained-from-scratch models; this robustness systematically improves with scale of both model and pretraining dataset size. We take initial steps towards characterizing what aspect of model representations allows them to perform continual learning so well, finding that in the pretrained models, distinct class representations grow more orthogonal with scale. Our results suggest that, when possible, scale and a diverse pretraining dataset can be useful ingredients in mitigating catastrophic forgetting.

Continual learning is a current challenge in designing machine learning systems: when trained on multiple tasks in sequence, models tend to suffer performance losses on earlier tasks—this is known as *catastrophic forgetting* (McCloskey & Cohen, 1989). Catastrophic forgetting occurs in many settings, from supervised learning for classification (Zenke et al., 2017) to reinforcement learning for games (Kirkpatrick et al., 2017) and downstream applications such as translation (Bapna & Firat, 2019) or medical diagnoses (Gupta et al., 2021). While many methods have been proposed to help mitigate forgetting, none of these have fully solved the problem; see Kemker et al. (2017) for a review.

Increasing model and dataset size have been key to the success of deep learning (Kolesnikov et al., 2020; BIG-bench collaboration, 2021; Radford et al., 2019; Zhai et al., 2021). A recent trend has been the use of increasingly large models pretrained on diverse datasets, which are subsequently fine-tuned or evaluated without fine-tuning on downstream tasks of interest; see Bommasani et al. (2021) and references therein. A particularly striking realization of this has been large pretrained language models, which show consistent improvement across scales (Kaplan et al., 2020) and can perform well on diverse tasks with no finetuning (Brown et al., 2020). From a continual learning perspective, this behavior is especially dramatic as, without finetuning, any training signal for these tasks appears mixed into a drastically larger pretraining dataset, and thus more than likely appeared many gradient steps before evaluation. This observation in large part inspired our work.

Here, we empirically study continual learning in large pretrained models. Our results suggest that pretrained models are significantly more resistant to catastrophic forgetting than their randomly-initialized counterparts, and that this resistance improves with the scale of both the model and pretraining data. Most of our experiments concern vision—specifically image classification; but we also conduct a preliminary investigation of language models.

Our key observations are as follows:

- Resistance to catastrophic forgetting improves with model and pretraining dataset scale (Figure 1).

- Large, pretrained image-classification models are significantly more resistant to forgetting than models trained from random initialization (Figure 6).

- Both supervised and unsupervised, SimCLR (Chen et al., 2020), pretraining improve forgetting (Supplement, Figure 12).

- We perform initial experiments aimed at exploring why scale and pretraining reduce catastrophic forgetting, showing that distinct class representations in pretrained models are more orthogonal than in trained-from-scratch models (Figure 9).

- We observe similar phenomena in language models (Supplement, Figure 11).

## 1  RELATED WORK

**Catastrophic forgetting** Catastrophic forgetting was identified as a problem for neural networks as early as the late 1980s, as it occurred in the single-hidden-layer networks that were studied at the time (see Delange et al. (2021) for a review). Within the catastrophic forgetting literature, a number of potential solutions to the problem have been devised; some involve replaying previous-task examples during training on later tasks (Rebuffi et al., 2016; Lopez-Paz & Ranzato, 2017; Chaudhry et al., 2018; Shin et al., 2017; Riemer et al., 2018), while others like elastic weight consolidation (Kirkpatrick et al., 2017) limit the extent to which parameters can change once learned (Zenke et al., 2017; Lee et al., 2017; Farajtabar et al., 2019).

Perhaps most similar to our work are proposed mitigation methods which exploit the overparameterization of modern neural networks to perform continual learning of multiple tasks, e.g. by using pruning in Mallya & Lazebnik (2018) or task-specific subnetworks in Wortsman et al. (2020) and recent works studying the impact of pretraining Anonymous (2022) and overparameterization via network width Mirzadeh et al. (2021) on forgetting.

**Large-scale models** The observation—initially in language models (Kaplan et al., 2020)—that model performance scales predictably with model and dataset size spurred an increased interest in training and deploying increasingly large pretrained models. Large-scale models emerged in natural-language processing with models such as GPT (Radford et al., 2019) and BERT (Devlin et al., 2019), which are pretrained in an unsupervised manner and thus able to make use of the large corpora of natural text available on the internet. More recently, image-classification models have also benefited from pretraining on large datasets (Kolesnikov et al., 2020), both supervised and unsupervised; a key example is the Vision Transformer architecture of Dosovitskiy et al. (2020), which—despite not using the inductive bias provided by convolutions—attains downstream performance comparable to convolutional networks after pretraining on ImageNet-21k (Deng et al., 2009) or JFT-300M (Sun et al., 2017). On many current NLP and computer vision benchmarks, large pretrained models achieve top performance.

All of the experiments in our work follow the pretrain-then-finetune method; recently, another paradigm, side-tuning, was proposed by Zhang et al. (2020), which avoids forgetting altogether by leaving the pretrained model fixed and employing auxiliary networks for downstream tasks.

## 2  EXPERIMENTAL SETUP

**Tasks:** Most of the experiments in this paper are conducted in the standard, task split setting (Kirkpatrick et al., 2017; Zenke et al., 2017). We consider *split CIFAR-10* task sequences – the ten class dataset is split and sequentially trained on two tasks of 5 classes each. We also consider sequential 10 and 50 class splits of CIFAR-100. Beyond split CIFAR, we study the following datasets: Oxford-IIIT pet, Oxford Flowers 102, Street View House Numbers (SVHN), Caltech-UCSD Birds 200 (CUB-200), Cars196, Domainnet/Clipart.

We also consider the more recent *input-distribution-shift CIFAR-100* setup (Ramasesh et al., 2021). In this case each task has the same target types made up a fixed subset of the 20 CIFAR-100 superclasses (e.g. aquatic mammals, fish, ...). The sequence of tasks consists of modifying which subclasses are sampled from. For example, Task A may have aquatic mammals drawn from beaver and dolphin, while Task B may have otter and whale. This setup allows one to probe forgetting without changing the model head, in a setting where task identity is not known to the model.

For our language model experiments, we consider next token prediction generative tasks and use the IMDb Reviews (Maas et al., 2011) and english Wikipedia (Foundation) datasets.

**Vision models:** In most of our experiments we use the Vision Transformer (ViT) and ResNet architectures. For the Vision Transformers, we use the ViT-S and ViT-B configurations from Dosovitskiy et al. (2020) and a smaller configuration we call ViT-xS – ranging from 5.7-86.7 million parameters. For ResNets, we use five models: ResNet26, ResNet50, ResNet101, ResNet152, and ResNet200 – ranging from 14.0-62.6 million parameters; as in Kolesnikov et al. (2020), these models use group normalization (Wu & He, 2018) and weight standarization (Qiao et al., 2019) in place of batch normalization. Details can be found in Tables 1 and 2 in Appendix A.1.2.

**Pretraining:** Unless otherwise specified, we pretrain our image models in a supervised manner on the ImageNet21k dataset, which contains approximately 14 million images in about 26000 distinct categories using the Adam optimizer (Kingma & Ba, 2017). Further details are in the Appendix.

**Finetuning:** When fine-tuning on multiple tasks in sequence, we use the SGD optimizer with momentum $\beta = 0.9$. The first task training employs a cosine decay schedule with a linear warmup; later tasks in the sequence are trained with fixed learning rates. For the split-CIFAR-10 task sequences, we use a multi-head setup in which each head is initialized to zeros.

## 3 RESULTS

### 3.1 FORGETTING IMPROVES WITH MODEL SCALE

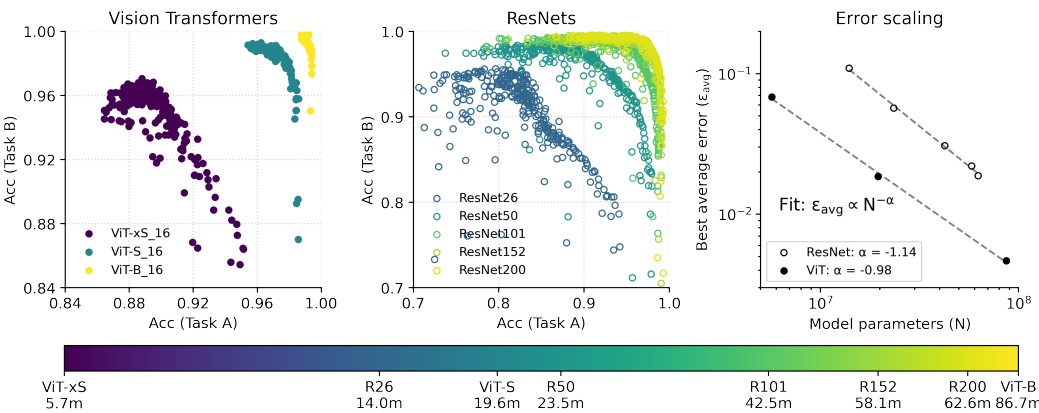

Figure 1: **Forgetting frontier across model scales**. Task A versus Task B performance for both pretrained vision transformers (left) and ResNets (center). Each point represents a different choice of learning rate or finetuning step for Task B. All models were pretrained on ImageNet21k for 90 epochs. (right) the best average Task A/B error improves systematically with model size.

Our main finding is that for both pretrained Vision Transformers and pretrained ResNets, catastrophic forgetting is mitigated to a large extent by scale: that is, larger models suffer less from forgetting. This is shown in Figure 1 for the two-task split CIFAR-10 sequence: Task A {airplane, automobile, bird, cat, deer} $\rightarrow$ Task B {dog, frog, horse, ship, truck}. In order to make a fair comparison, we plot the test accuracies of the model on both tasks at different steps of Task B fine-tuning for a variety of learning rates. This allows us to see, for each model, the maximum performance achievable on the new task for a given performance on the initial task. We call this the forgetting frontier.

Figure 1 shows that the forgetting frontier for both ViT and ResNet models improves significantly with scale. While a 5.7M-parameter ViT-xS model loses about 6.5% in Task A accuracy over the course of learning to perform Task B, an 86.7M-parameter ViT-B model loses less than half a percent. To put this in context, in a previous studies of split-CIFAR-10 task sequences, a ResNet18 model lost close to 40% accuracy with no mitigation methods, 5% when using a replay buffer, and 20% when using elastic weight consolidation (see e.g. Figure 1 of Ramasesh et al. (2021)).

It is also important to note that this effect of scale and pretraining on improving forgetting frontiers does not seem to be architecture-dependent. Both pretrained vision transformers—based on self-

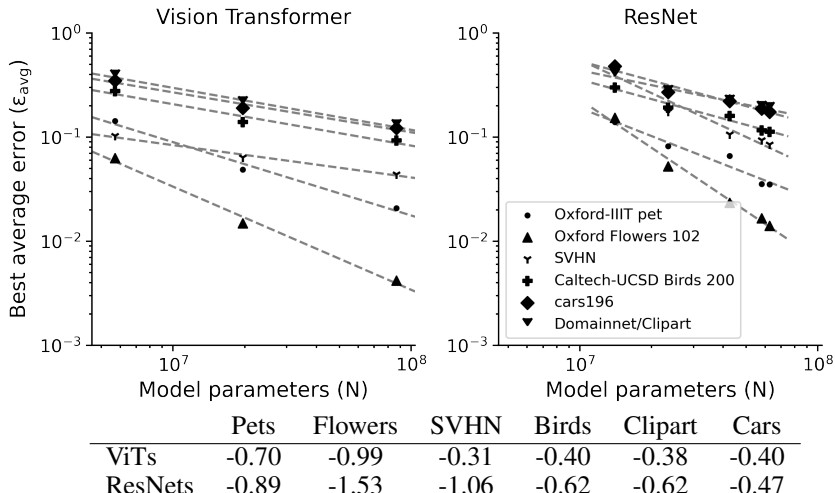

| | Pets | Flowers | SVHN | Birds | Clipart | Cars |
|---|---|---|---|---|---|---|
| ViTs | -0.70 | -0.99 | -0.31 | -0.40 | -0.38 | -0.40 |
| ResNets | -0.89 | -1.53 | -1.06 | -0.62 | -0.62 | -0.47 |

Figure 5: **Scaling behavior on non-CIFAR, two-task sequences**. (top) While the rate of improvement varies across datasets, continual-learning performance in these two-task continual learning setups systematically improves with model size, consistent with a power-law. For a full discussion and plot of all forgetting frontiers, see Figure 2. (bottom) Table of power-law fit exponents.

attention—and ResNets—based on convolutions—exhibit improvements in their forgetting frontiers as they grow larger.

We see this improvement in forgetting performance beyond the simple two-task split-CIFAR-10 scenario. Across a 10-task sequence of 10-class split-CIFAR-100 tasks (Figure 3 left), sequential 50-class splits of CIFAR-100 (Figure 3 right), the CIFAR-100 single-head input-distribution-shift task (Figure 4), and two-task sequences beyond the CIFAR dataset (Figure 2), large-scale pretrained models exhibit resistance to forgetting, and continual learning ability appears to improve systematically with scale. For detailed results, scaling plots, and forgetting frontiers in the 10-task scenario, see Section B.3 in the Supplement.

## 3.2 DEPENDENCE ON FINETUNING DATASET

It is interesting to study how the improved resistance to forgetting with scale depends on the choice of finetuning dataset. In particular, it is worth noting that, though quite different in format, CIFAR-10/100 are semantically quite similar to subsets of ImageNet21k, our pretraining set. To empirically investigate the dependence on finetuning dataset we have run additional experiments similar to those presented in Figure 1 for the Oxford Pet and Flowers, Caltech Birds, Stanford Cars, street view house numbers (SVHN) and DomainNet Clipart datasets. Results are shown in Figure 5 and in Figure 2. We note in particular that SVHN classes are numerals and so distinct from ImageNet21k classes, while the Clipart dataset consists of synthetic, rather than natural images. See the Supplement for further details.

## 3.3 FORGETTING IN PRETRAINED MODELS VS. MODELS TRAINED FROM SCRATCH

In the previous section, we saw that the amount of catastrophic forgetting in models pretrained on large datasets decreased with increasing model size, to the point that the largest models (ViT-B, with 86.7M parameters and ResNet200 with 62.6M parameters) suffer very little performance loss from second-task training. In this section, we investigate whether this improvement in forgetting is simply a consequence of the improvement in performance on Task A. That is, do pretraining and scale have benefits with respect to forgetting beyond simply improving the fine-tuned performance on downstream tasks?

To check this, we compare the performance and amount of forgetting in pretrained ResNet models to models trained from scratch. We handicap the pretrained ResNets so that their maximum accuracy on on Task A matches that of the corresponding trained-from-scratch ResNet; this is done simply by

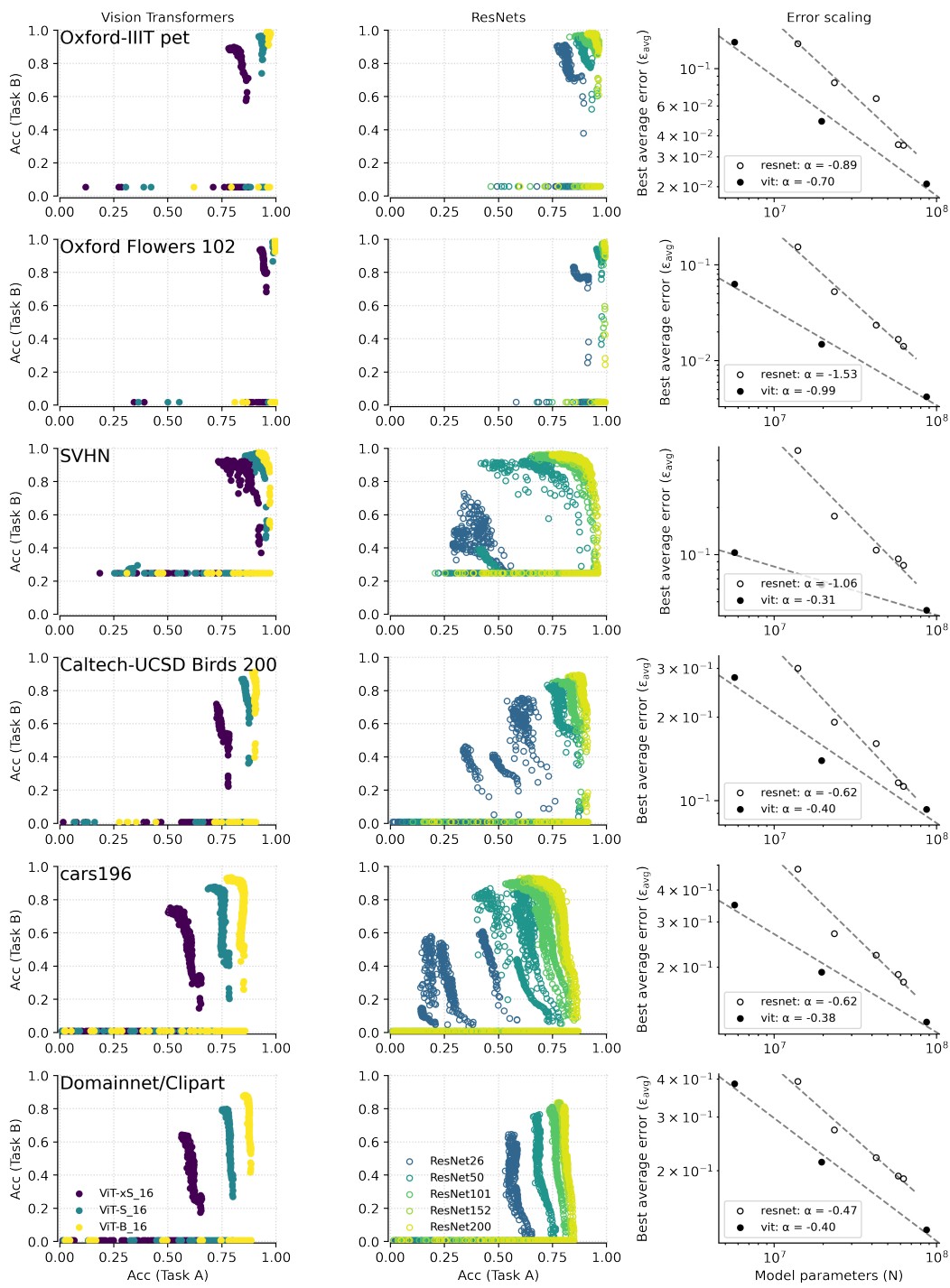

Figure 2: **Forgetting frontiers for two-task sequences beyond CIFAR10 and CIFAR100**. Fine-tuning settings for these experiments are shown in table 4 (supplement), while the datasets themselves are descrbed in section B.4 (supplement). As with the CIFAR10 and CIFAR100 datasets, we find that model scale is beneficial from the perspective of continual learning, though especially for smaller ResNets the behavior is less clean than on CIFAR. Scaling exponents also vary with dataset.

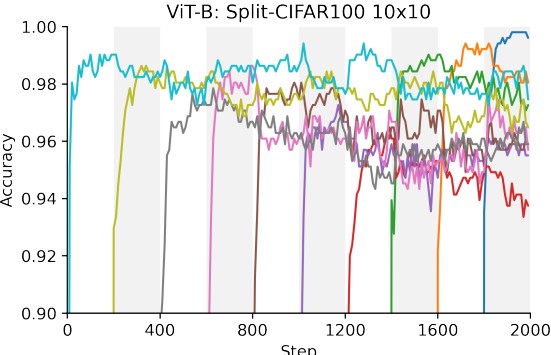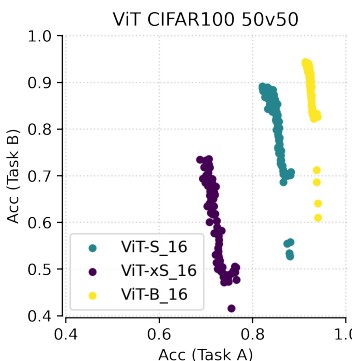

Figure 3: **Split CIFAR-100 tasks**. (left) Sequential training of a pretrained ViT-B model on a 10-task sequence, where each task consists of classifying between 10 categories in the CIFAR-100 dataset. In this continual learning setup, the model is able to learn all 10 tasks without much forgetting: on average, the model loses 1.8% accuracy on each task; at most, the model loses 2.9%. (right) Forgetting frontiers for vision transformers on a 2-task, 50-class split CIFAR-100, showing the same increase with scale as on the split-CIFAR-10 version.

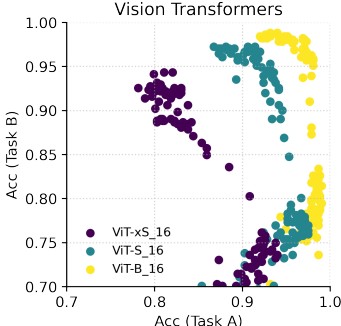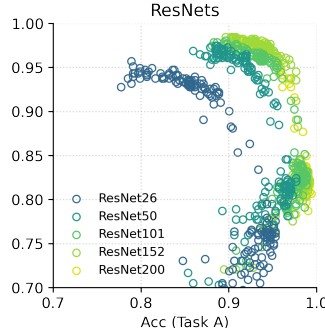

Figure 4: **Single-head models on CIFAR-100 distribution shift task**. As with the multi-head setup, the forgetting in Vision Transformers (left) and ResNets (right) is largely mitigated by pretraining and scale. The full task details are given in Appendix A.1.3

reducing the number of steps for which we fine-tune. Figure 6 shows the results of this experiment, with additional results and details in Appendix C.1. While the trained-from-scratch ResNets achieve the same performance as their pretrained counterparts on Task A, their Task A performance degrades significantly more than the pretrained ResNets when trained on Task B; pretrained ResNets forget less, and this improves with model scale.

This result suggests that pretraining and model scale confer benefits upon models which go beyond simply improving performance. From Figure 6 it is also clear (see also Figure 19 in the appendix, which plots the same data in a different way) that pretraining is required to see the benefits of scale. Simply scaling up the model size in trained-from-scratch ResNets does not yield clear improvements in their forgetting performance.

We focused in this section exclusively on ResNets (rather than Vision Transformers) because obtaining decent performance on image classification tasks with Vision Transformers seems to require pretraining. In our experiments, for example, a trained-from-scratch ViT-S model was unable to achieve better than 75% test accuracy on five-class CIFAR-10, while the same model, pretrained, achieved better than 98% accuracy.

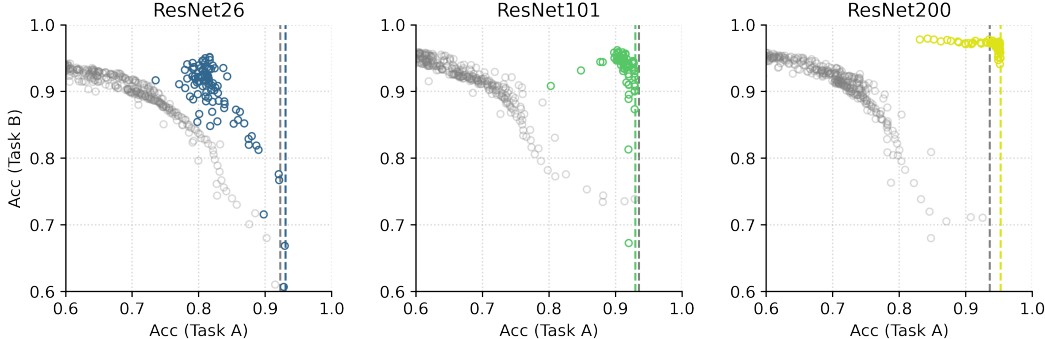

Figure 6: **Pretrained versus trained-from-scratch models**. ResNet models trained from scratch (gray) exhibit inferior forgetting performance than pretrained models (colored), even for the pretrained models shown here, which were handicapped during fine-tuning, in order to match Task A performance. Futhermore, the trained-from-scratch models are unable to take advantage of model scale – the forgetting frontier is largely independent of model size (see also Figure 19 in the Appendices). Dashed vertical lines show the maximum accuracy on Task A.

## 3.4 DEPENDENCE ON PRETRAINING TIME, PRETRAINING DATASET SIZE, AND FINETUNING TIME

Having seen in the previous sections that pretraining on large datasets improves models' resilience to catastrophic forgetting, in this section we explore the dependence of this phenomenon on the duration of pretraining, the size of the pretraining dataset, and the number of steps for which we finetune on Task A.

First, we study the influence of pretraining time on downstream forgetting. This is motivated by previous studies (e.g. Hernandez et al. (2021)) showing that downstream performance does not always monotonically improve with pretraining time and performance. By contrast, in the case of forgetting we find (see Figure 7 for ViT-S results) that as the pretraining process continues, both downstream performance and the forgetting frontier improve. This was the case for all models (ViTs and ResNets) and downstream task sequences (CIFAR-10, CIFAR-100, etc.) we studied. We found no evidence of forgetting frontiers getting worse with increased pretraining.

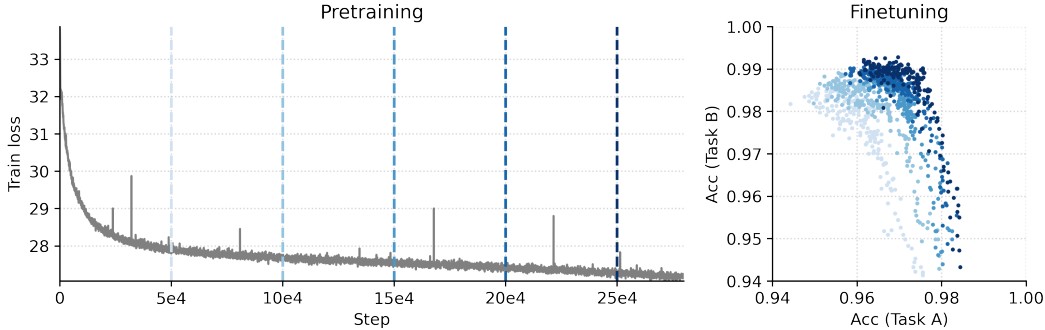

Figure 7: **Varying pretraining time**. We take pretrained checkpoints at 50k, 100k, 150k, 200k, and 250k steps (shown in colored ticks at the left) for a ViT-S model pretrained on ImageNet21k. (right) The forgetting frontiers improve with increased pretraining time.

Along similar lines, we show the influence of pretraining data scale on downstream forgetting in Figure 8, pretraining a model on varying fractions of ImageNet21k. While the full dataset contains over 14 million images, we find that training on even one-sixteenth of the dataset (roughly the size of the standard ImageNet ILSVRC 2012 dataset) yields a model which only loses about 3% performance in forgetting on split-CIFAR-10. This suggests that one does not need to pretrain models on unreasonably large datasets in order to reap benefits for continual learning.

Finally, we study the forgetting frontier of pretrained models as we vary the amount of first-task fine-tuning (number of training steps). This is partly inspired by a recent study (Andreassen et al., 2021) of robustness in pretrained models, which found that during fine-tuning, pretrained models evolved some robustness to distribution shift which vanished at the end of fine-tuning. Figure 8 shows forgetting frontiers for pretrained ViT-B and ResNet101 models, fine-tuned for a number of steps between 100 and 5000. The results in the figure suggest that in our setting, increasing the fine-tuning steps to maximize performance on Task A does not appear to cause increased forgetting. On the contrary, even training for about ten times more steps than it takes to achieve saturating performance does not impact forgetting.

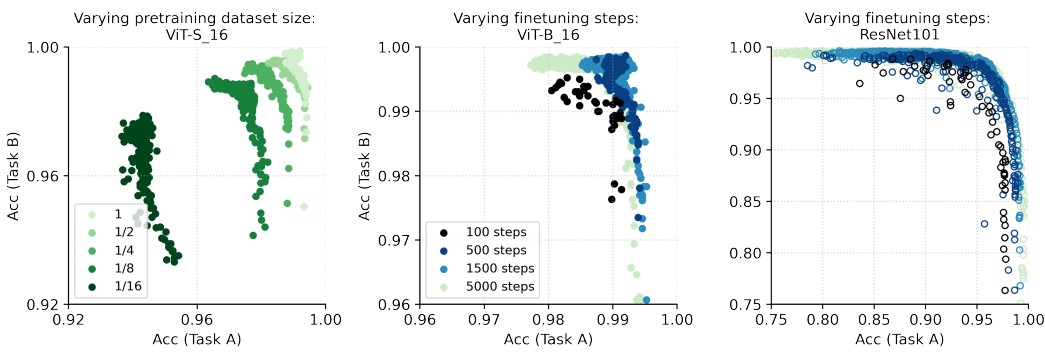

Figure 8: **Varying dataset size and finetuning time**. (left) The forgetting frontier for a vision transformer (ViT-S16) finetuned on a two-task CIFAR-10 sequence, pretrained on varying fractions of the ImageNet21k dataset shows improvement with pretraining dataset size. (center, right): The forgetting frontiers for models pretrained on ImageNet21k, with varying amounts of fine-tuning – Task A performance saturates around 500 steps; even training for ten times as many steps does not affect the forgetting frontier.

## 3.5 REPRESENTATION OVERLAP AND FORGETTING

In this section we take steps toward understanding why large pretrained models seem to be resistant to forgetting. Following the work of Ramasesh et al. (2021), we measure the similarity between a models representation of Task A and Task B data.

In detail, if we denote the $P$ model features (penultimate layer activations) on an input, $x$, by $f(x) \in \mathbb{R}^P$, Ramasesh et al. (2021) propose measuring the similarity of $f_A := \{f(x) : x \in \text{Task A}\} \in \mathbb{R}^{|\text{Task A}| \times P}$ and $f_B := \{f(x) : x \in \text{Task B}\} \in \mathbb{R}^{|\text{Task B}| \times P}$ using the *trace overlap*,

$$S_{AB} = \frac{\text{Tr}(\Theta_{AB}\Theta_{AB}^{\text{T}})}{\sqrt{\left(\text{Tr}(\Theta_{AA}\Theta_{AA}^{\text{T}})\text{Tr}(\Theta_{BB}\Theta_{BB}^{\text{T}})\right)}} . \tag{1}$$

Here $\Theta_{AB} = f_A f_B^T$ is the matrix of inner-products between Task A and Task B features.

Ramasesh et al. (2021) show empirically that forgetting is maximal for intermediate values of $S_{AB}$, and derive this property analytically in a simplified model. In particular Ramasesh et al. (2021) propose that a minimal trace overlap (corresponding to orthogonality of representations of different tasks) will lead to minimal forgetting. Inspired by this, we measure representation overlaps in models finetuned on split CIFAR-10, taking representations at the end of Task A training.

We consider the 10x10 matrix, $S$, formed by taking the indices $A$ and $B$ to run over the different classes in CIFAR-10. The numerator in the above expression can be thought of as an empirical (batch) estimate of the inner product between feature vectors for random draws from classes $A$ and $B$; trace overlaps close to zero indicate that representations are nearly orthogonal.

Using the above definition, we return to the original split-CIFAR-10 setting of §3.1 and §3.3. For each of the models used in those experiments (see Tables 1 and 2), we compute the representation overlap matrix. We focus on two comparisons: (i) how does the representation overlap matrix differ between pretrained and trained-from-scratch ResNets? and (ii) how does the orthogonality of representations vary as we scale the model size, for both ResNets and Vision Transformers?

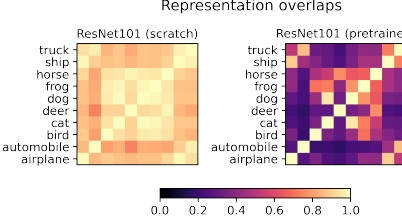
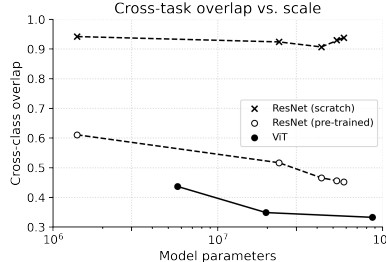

Figure 9: **Trace overlap of CIFAR-10 class representations**. (left) The cross-class representation overlap matrices (Equation 1) are significantly lower for a pretrained ResNet101 than for one trained from scratch. (right) For both Vision Transformers and ResNets, the cross-class overlap decreases as the model size increases. These representation overlaps are computed *after* training the model on the initial CIFAR-10 subtask. See Appendices for additional models and measurements.

**Pretrained vs. trained-from-scratch representations**: We show representative trace overlap matrices for both pretrained and trained-from-scratch ResNet101 models in Figure 9(left and center) (see Figure 27 for additional models). Each element of this matrix corresponds to the representation overlap between two batches, each drawn entirely from a single class. The representation overlap between disjoint classes (off-diagonal class overlap matrix elements) is much smaller for the pretrained ResNets than those trained from scratch. Averaging cross-class overlaps over class pairs from different tasks quantifies this difference: for trained-from-scratch ResNet101, the average overlap is $0.907$, while for the pretrained model this value is only $0.466$. For all sizes of ResNet, the pretrained models exhibit representations which have much smaller overlap than their trained-from-scratch counterparts.

**Representation overlap across scales**—Figure 9(right) shows the average cross-task overlap for both ResNets and Vision Transformers plotted against model size (number of parameters). For pretrained Vision Transformers and ResNets, this overlap decreases with model size, while for trained-from-scratch ResNets, there is not a clear decreasing trend. Interestingly, Vision Transformers appear to form more orthogonal representations than ResNets do, a fact which can also be seen in the overlap matrices presented in supplementary Figure 26, and perhaps explains their improved forgetting performance.

These observations suggest that pretrained models store representations of different classes with much less overlap than trained-from-scratch models, and that this orthogonality increases with scale. As this mirrors the the observed trend in forgetting in these models, it is suggestive that the increased orthogonality of representations is a partial explanation for the forgetting resilience seen in large pretrained models.

## 4 DISCUSSION

The central observation we have reported in this paper is that scaling up models pretrained on large datasets is effective at mitigating catastrophic forgetting in the settings studied. Our experiments show that pretraining and scale can confer benefits beyond simply the achievable downstream loss or accuracy values.

An important direction for future work is understanding what underlies the systematic improvement of continual learning with model scale. The rate at which performance improves with model size in Figure 1, with power-law exponent close to $-1$, is reminiscent of the *variance limited* behavior discussed in Bahri et al. (2021). Namely, when the number of model parameters $N$ is much larger than the number of training examples (as is the case for split-CIFAR finetuning), performance is predicted to scale as $N^{-1}$. This scaling behavior is notably *not* seen in Figure 11 for language modeling—this is reminiscent of the disparate scaling behaviors observed in Sharma & Kaplan (2020) between language-modeling and image-classification tasks. Ideally, developing an understanding of this behavior would allow for predicting the rate of improvement to forgetting with model size.

ACKNOWLEDGMENTS

The authors would like to thank Anselm Levskaya, Bill Mark, Anders Andreassen, and Maithra Raghu for their conversations during the completion of this work.

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

## A EXPERIMENTAL DETAILS

### A.1 VISION

#### A.1.1 SUPERVISED PRETRAINING

Here we detail the settings used to in supervised pretraining, resulting in the models used in the experiments of sections 3.1, 3.3, 3.4, and B.2. We pretrain seven models: (ViT-B, ViT-S, ViT-xS, R26x1, R50x1, R101x1, R152x1, R200x1) on ImageNet21k for 90 epochs. Pretraining is done using the Adam optimizer ($\beta_1 = 0.9$ and $\beta_2 = 0.999$); for all models, we use a batch size of 4096. Our learning-rate schedule includes a warmup of 10k steps to a maximum learning rate $\eta = 10^{-3}$, followed by a linear decay to $10^{-5}$. For the Vision Transformer models, we use a weight decay penalty of $0.03$ and a dropout rate of $0.1$; for ResNet models, we do not use weight decay or dropout. These settings are largely identical to those found in Dosovitskiy et al. (2020).

A note on ImageNet21k: We use the version of ImageNet21k available in TensorFlow Datasets 4.0.1 and apply minimal preprocessing, simply converting the labels to one-hot vectors and cropping the image to be 224x224 pixels in size. During training, this crop is an Inception-style random crop, while during evaluation it is a deterministic central crop; during training we additionally augment the data by performing random horizontal flips and apply label smoothing, with values 0.9999 and 0.0001. Importantly, ImageNet21k is class-imbalanced and heavy-tailed, with the uneven distribution of label occurrences shown in Figure 10. We do not balance the dataset.

#### A.1.2 VISION TRANSFORMER MODEL DETAILS

For full description of the Vision Transformer architecture, see the original paper by Dosovitskiy et al. (2020). Abbreviated details are given here. We process 2D images of height $H$, width $W$, and $C$ channels, $\mathbf{x} \in \mathbb{R}^{H \times W \times C}$ by first reshaping them into a 1D sequence of flattened patches–in this work, we exclusively use patch size 16. Each patch is mapped (via a learned linear transformation) to an embedding vector of dimension $D$ (for our models, $D_{xS} = 256$, $D_S = 516$, and $D_B = 768$, as in Table 1). As both our pretraining and finetuning is supervised, we prepend a (learnable) '[class]' token embedding to this sequence of patches, add a learnable 1D position embedding, and pass the sequence through the Transformer encoder. The Transformer encoder alternates between multi-headed self-attention layers and MLP (plus residual connection) layers, with layernorm applied before both the attention and MLP layers.

For classification, we use a classification head which acts on only the transformed '[class]' token representation. During pretraining, this classification head is an single-hidden-layer MLP, as in the original Vision Transformer work; during finetuning, the classification head is a linear layer (in the split-CIFAR-10 setting, we use a multi-head setup with a different classification layer for each task in the sequence). We initialize all classification layers with zeros.

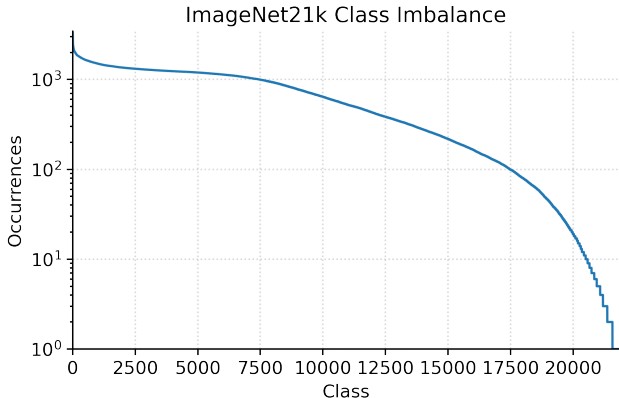

Figure 10: **ImageNet21k is highly class-imbalanced**. While roughly 7500 classes of the ImageNet21k dataset feature over 1000 examples, there are over 3000 classes with less than 100 examples. As mentioned in the main text, in this work we do not class-balance ImageNet21k before pretraining.

| Model | Layers | Width | Dim. | Heads | Params | |
|---|---|---|---|---|---|---|
| ViT-xS | 4 | 256 | 256 | 4 | 5.7M | |
| ViT-S | 6 | 516 | 516 | 6 | 19.6M | |
| ViT-B | 12 | 768 | 768 | 12 | 86.7M | |

| Model | Resblocks | Params | |
|---|---|---|---|
| R26 | [2, 2, 2, 2] | 14.0M | |
| R50 | [3, 4, 6, 3] | 23.5M | |
| R101 | [3, 4, 23, 3] | 42.5M | |
| R152 | [3, 8, 36, 3] | 58.1M | |
| R200 | [3, 24,36,3] | 62.6M | |

Table 2: Configurations of ResNet models

### A.1.3 CIFAR-100 DISTRIBUTION SHIFT TASK

Here we provide more information about the CIFAR-100 distribution-shift task, which was introduced in Ramasesh et al. (2021) and forms the basis of our experiments in section A.1.3 of the main text. The idea behind this task is to mimic a way in which catastrophic forgetting might occur in practice, in image-classification settings where the class labels remain constant but the distribution of the images changes. A key features of this type of setting is that the identity of the task is not known to the model at inference time, so task-specific components (such as multiple readout layers) are not allowed. For example, in a medical setting, one might seek to classify images as displaying signs of a disease or not, and the system might be trained sequentially on images taken at various hospitals. While the CIFAR-100 dataset features images divided into 100 distinct classes, these classes are grouped into 20 *superclasses*. The CIFAR-100 distribution shift task sequence involves classifying an image by its superclass; individual tasks in the sequence draw images from a single subclass for each superclass. As a concrete example, in the experiments in the main text we use the five superclasses *aquatic mammals*, *fruits and vegetables*, *household electrical devices*, *trees*, and *vehicles-1*[1]. For the first task in the sequence, Task A, we uses the classes dolphin (for aquatic mammals), apple (for fruits and vegetables), lamp (for household electrical devices), maple tree (for trees), and bicycle (for vehicles-1). For the second task, Task B, we use the classes whale (for aquatic mammals), orange (for fruits and vegetables), television (for household electrical devices), willow (for trees), and motorcycle (for vehicles-2).

### A.2 FINETUNING EXPERIMENTAL DETAILS

In this section we provide details on the finetuning procedures for the experiments in the main text. All finetuning was done using stochastic gradient descent with momentum ($\beta = 0.9$), clipping gradients at unity; batch size is fixed at 512. In the first task, we used a schedule with a 5-step linear

---

[1]The CIFAR-100 dataset features two vehicle superclasses, denoted *vehicles-1* and *vehicles-2*

ramp to a maximum learning rate followed by a cosine decay to zero over the full first-task training. In the second task, we used constant learning rates.

**Section 3.1, Figure 1:** Finetuning both the Vision Transformers and ResNets was done for 500 steps. For Vision Transformers, the first-task training used a warm-up from $\eta = 0.01$ to $\eta = 0.05$. For each model we perform four different runs, with second-task (constant) learning rates 0.001, 0.003, 0.007, and 0.01. For ResNets, the task-A learning rate warm-up was from $\eta = 0.002$ to $\eta = 0.01$. The (constant) second-task learning rates we used were 0.008, 0.004, 0.002, 0.001, 0.0003, and 0.0001.

**Section 3.1, Figure 2:** In the 10-task split-CIFAR100 sequence (left), each task was trained for 200 steps; the initial task used a learning-rate schedule with a 5-step warmup from $\eta = 0.01$ to $\eta = 0.05$ followed by a cosine decay to 0 over the 200 steps, while all subsequent steps used constant $\eta = 0.005$. In the 2-task 50-class split-CIFAR100 sequence (right), the settings are identical to those of the Vision Transformer training in Figure 1.

**Section 3.1, Figure 3:** For ResNets, we used a first-task warmup from $\eta = 0.001$ to $\eta = 0.005$, and trained each task for 300 steps. Constant second-task learning rates were 0.001, 0.0003, and 0.0001. For Vision Transformers, we trained each task for 200 steps, using a first-task warmup from $\eta = 0.01$ to $\eta = 0.05$. Constant second-task learning rates were 0.01, 0.003, and 0.001.

**Section 3.2, Figure 4:** The pretrained models in this figure were the same as those in Figure 1. For the trained-from-scratch models, we used a linear warm-up from $\eta = 0.006$ to $\eta = 0.03$, and trained each task for 15,000 steps. Constant second-task learning rates were 0.001, 0.0003, and 0.0001. Note that training here is much longer than for the pretrained models.

**Section 3.3, Figure 5:** Vision Transformer training settings are identical to those of Figure 1.

**Section 3.3, Figure 6:** In the left panel, Vision Transformer training settings are identical to those of Figure 1. In the right two panels, settings are also identical to those of Figure 1, with the exception of the number of finetuning steps, as indicated in the figure legend.

**Section 3.4, Figure 7:** We finetune for 300 steps on each task, with a maximum learning rate of $\eta = 0.0002$ (we omit the linear warm-up for these models, but keep the cosine decay). Constant second-task learning rates are $\eta = 0.0002$, $\eta = 0.0001$, $\eta = 0.00004$, and $\eta = 0.00002$.

## A.3 LANGUAGE

Models are decoder-only transformers which were pretrained using Adafactor and learning rate $1.0$, context length $1024$ and batch size $256$. We finetuned with batch size of $8$ and set a constant learning rate equal to the learning rate at the end of pretraining rescaled by the relative batch size, $8/256$.

## B ADDITIONAL EXPERIMENTS

### B.1 FORGETTING IN LANGUAGE MODELS

Transfer learning from unsupervised pretraining is commonplace in Natural Language Processing. It has become standard to pretrain models on data crawled from the internet and then finetune on smaller, downstream tasks (Devlin et al., 2019; Radford et al., 2019; Raffel et al., 2020; Brown et al., 2020). This bears similarities with both the unsupervised pretraining and the experiments in Figure 4 where a single head was used to train on different distributions (subclasses). Indeed, as mentioned in the main text, part of the conceptual motivation for this work came from the observation that large language models with diverse pretraining perform well on tasks with no finetuning – ostensibly retaining information for many gradient steps. Here, we include some preliminary empirical results probing catastrophic forgetting directly in large language models.

We consider decoder only transformer models pretrained on data scraped from the web, including C4 (Raffel et al., 2020). These models range in size from 57 million to 8.6 billion non-embedding parameters and were originally trained as part of the BIG-bench collaboration (2021). In Figure 11, we consider sequentially finetuning the models on Task A: IMDb Reviews for 1.5k steps followed by Task B: english Wikipedia for 10k steps. The longer finetuning time for Task B is intended to highlight forgetting, as these models are relatively robust.

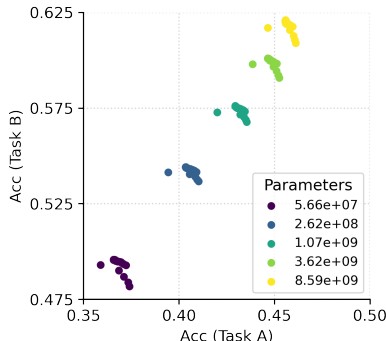 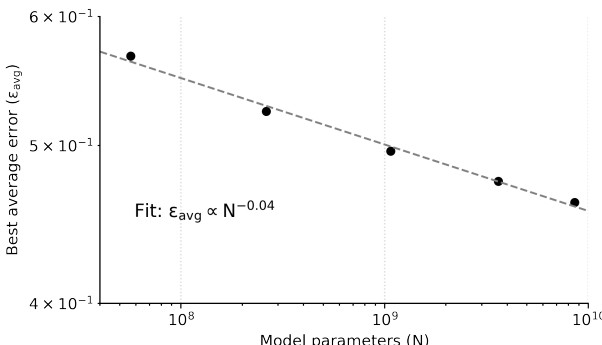

Figure 11: **Forgetting in sequential language-modeling**. Token accuracy for decoder-only transformer models show systematic improvement in joint Task A, Task B performance across scales.

We find that, as in the case of pretrained vision models, the best joint Task A, Task B performance improves with scale.

## B.2 FORGETTING IN VISION MODELS WITH UNSUPERVISED PRETRAINING

In the main text, the models we studied were pretrained in a supervised manner on ImageNet21k. However, part of the conceptual motivation for this work came from the properties of language models trained in an unsupervised fashion. Furthermore, recently, unsupervised pretraining methods have yielded vision models which achieve good downstream performance; among these are SimCLR (Chen et al., 2020), MoCo (He et al., 2020), DINO (Caron et al., 2021), and BYOL (Grill et al., 2020). Given the increased prevalence of such unsupervised pretraining methods, we investigate in this section whether unsupervised pretraining, combined with scale, provides resistance to forgetting similarly to supervised pretraining. We take publicly-available ResNet models which were pretrained using SimCLR on the ImageNet ILSVRC-2012 dataset (Russakovsky et al., 2015), and finetune them on the two-task split CIFAR-10 sequence. The resulting forgetting frontiers, shown in Figure 12, show that as in supervised pretraining, SimCLR pretraining yields models which are resilient to forgetting, especially as scale is increased. We leave a thorough exploration of the continual-learning properties of other unsupervised pretraining methods for future work.

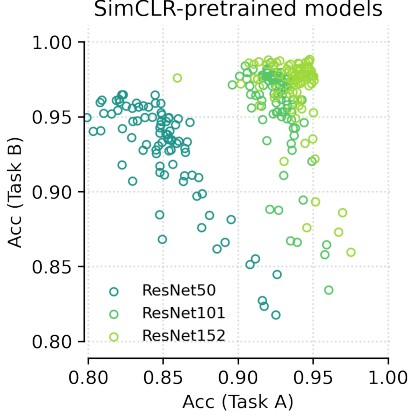

Figure 12: **Forgetting frontiers on a split-CIFAR10 task for models pretrained in an unsupervised manner on ImageNet1k**. ResNets pretrained using unsupervised SimCLR on ImageNet1k exhibit a similar trend to that observed in the supervised setting of forgetting robustness improving with model scale.

## B.3 FORGETTING IN TASK SEQUENCES WITH MORE THAN TWO SUBTASKS

In this section, we extend our results on vision past the two-task setting which was the primary focus of the main text. Specifically, we use a 10-class, 10-subtask split of the CIFAR100 dataset to study how forgetting in this scenario varies as a function of model and dataset scale. The specific split we use is shown in Table 3. As in the main text, we study the performance of Vision Transformer and ResNet models.

| Task A | Task B | Task C | Task D | Task E |
|---|---|---|---|---|
| Apple | Bowl | Chair | Dolphin | Lamp |
| Aquarium Fish | Boy | Chimpanzee | Elephant | Lawnmower |
| Baby | Bridge | Clock | Flatfish | Leopard |
| Bear | Bus | Cloud | Forest | Lion |
| Beaver | Butterfly | Cockroach | Fox | Lizard |
| Bed | Camel | Couch | Girl | Lobster |
| Bee | Can | Crab | Hamster | Man |
| Beetle | Castle | Crocodile | House | Maple tree |
| Bicycle | Caterpillar | Cup | Kangaroo | Motorcycle |
| Bottle | Cattle | Dinosaur | Keyboard | Mountain |

| Task F | Task G | Task H | Task I | Task J |
|---|---|---|---|---|
| Mouse | Plain | Rose | Squirrel | Train |
| Mushroom | Plate | Sea | Streetcar | Trout |
| Oak tree | Poppy | Seal | Sunflower | Tulip |
| Orange | Porcupine | Shark | Sweet pepper | Turtle |
| Orchid | Possum | Shrew | Table | Wardrobe |
| Otter | Rabbit | Skunk | Tank | Whale |
| Palm tree | Raccoon | Skyscraper | Telephone | Willow tree |
| Pear | Ray | Snail | Television | Wolf |
| Pickup truck | Road | Snake | Tiger | Woman |
| Pine tree | Rocket | Spider | Tractor | Worm |

Table 3: 10-subtask split of CIFAR100 used in our experiments.

We find that:

- As in the two-task setting, both the forgetting frontiers and the best average accuracy improve with model scale as a power-law in the number of model parameters.
- When we scale the size of the pre-training dataset, we also observe an improvement with scale, though with some saturation around the full ImageNet21k dataset..

**Model scaling:** Model-scaling plots showing the best average accuracy as a function of model parameters are show in Figure 13. As in the two-task split CIFAR10 sequence, these scaling plots are one way to show that the continual-learning performance of both architectures improves with scale. The power-law exponents are different in this 10-task setting than they are for the two-task setting, but interestingly all are close to unity.

Another way to see the improvement in continual-learning ability with model scale is to plot forgetting frontiers, which we do in Figures 15 and 16. In this setting with more than two tasks, plotting a single forgetting frontier is not possible, but we plot two-task frontiers for all pairs of subtasks (with 10 subtasks, there are $10 \times 9/2 = 45$ pairs). These frontiers are constructed in exactly the same way as the two-task frontiers, showing the test accuracies for each subtask throughout the entirety of the 10-task training. As the figures show, forgetting frontiers of larger models dominate the forgetting frontiers of smaller models of the same architecture. This complements the previous scaling-plot as a demonstration of improving continual-learning with scale, though the behavior here is not as clean as it was in the two-task setting.

**Data scaling:** A data-scaling plot showing the best average accuracy for a ViT-S model as a function of pretraining dataset size is shown in Figure 14. Forgetting frontiers are shown in Figure 17.

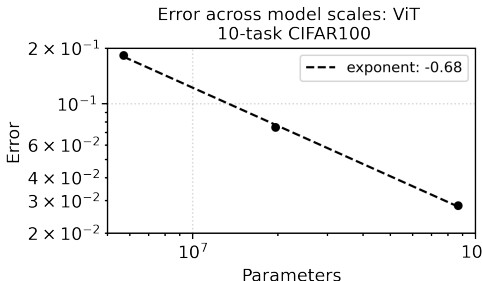 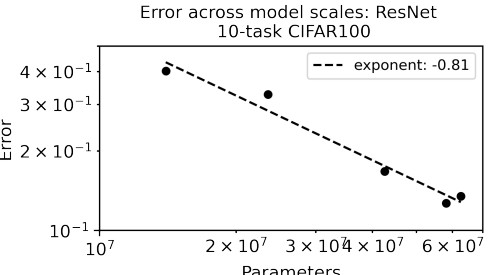

Figure 13: **Model scaling on 10-subtask split-CIFAR100**. (left) Scaling of best average error with parameter count for vision transformer models. (right) Scaling of best average error with parameter count for ResNet models.

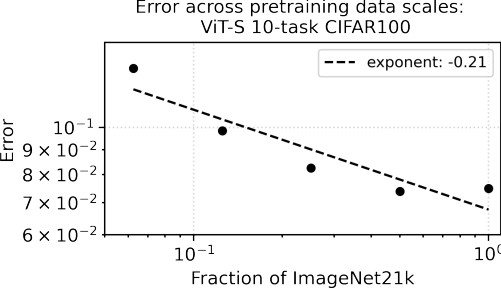

Figure 14: **Data scaling on 10-subtask split-CIFAR100**. Scaling of best average error with pre-training dataset size (measured in a fraction of the full ImageNet21k dataset) for the ViT-S_16 model.

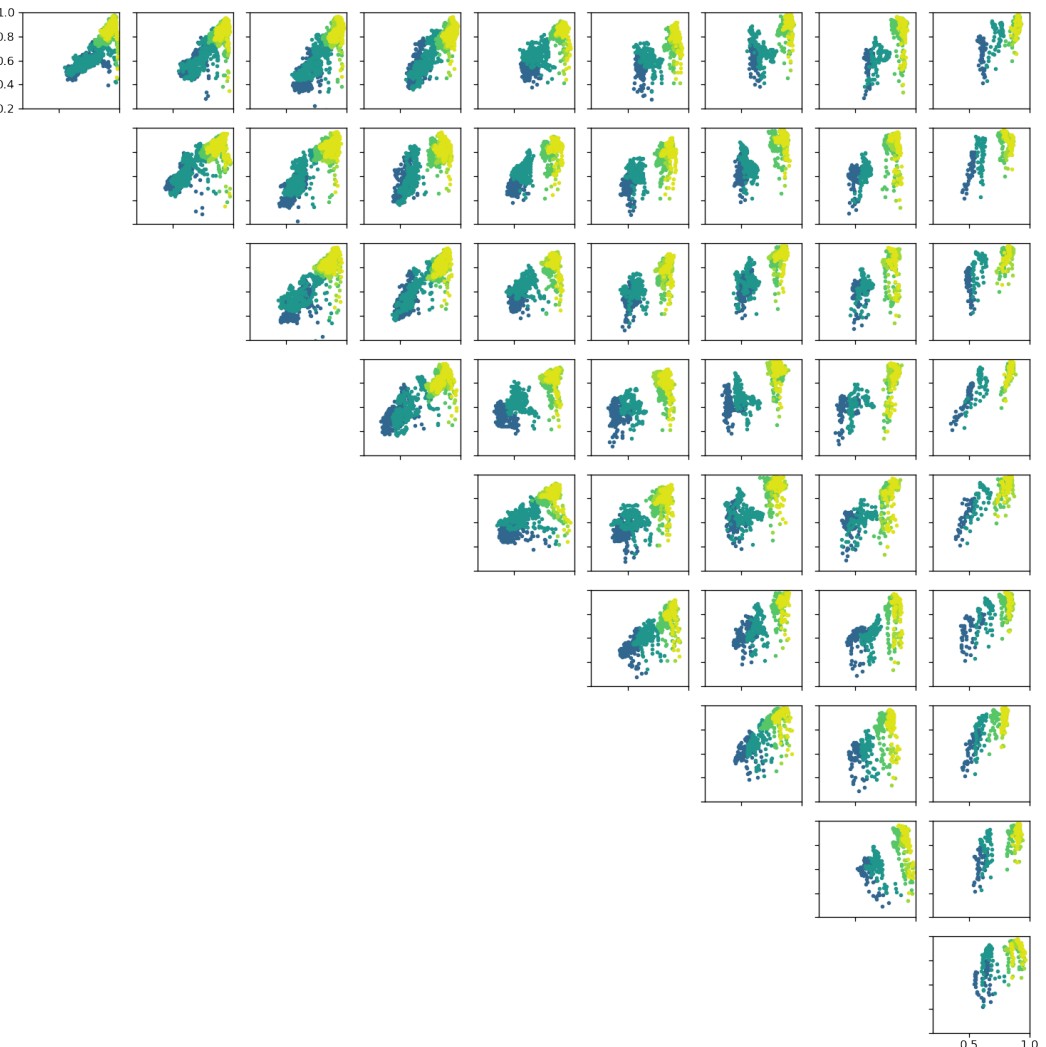

Figure 15: **Forgetting frontiers for 10-task Split CIFAR100, Resnet**. Colors are the same as in Figure 1 of the main text. All images in each row share the same task for the x-axis, i.e. the x-axis of the top row represents task A, x-axis of the second-from-the-top row represents task B, etc. All images in each column share the same task for y-axis, i.e. the y-axis of the leftmost column represents task B, the y-axis of the second-to-leftmost column represents task C, etc.

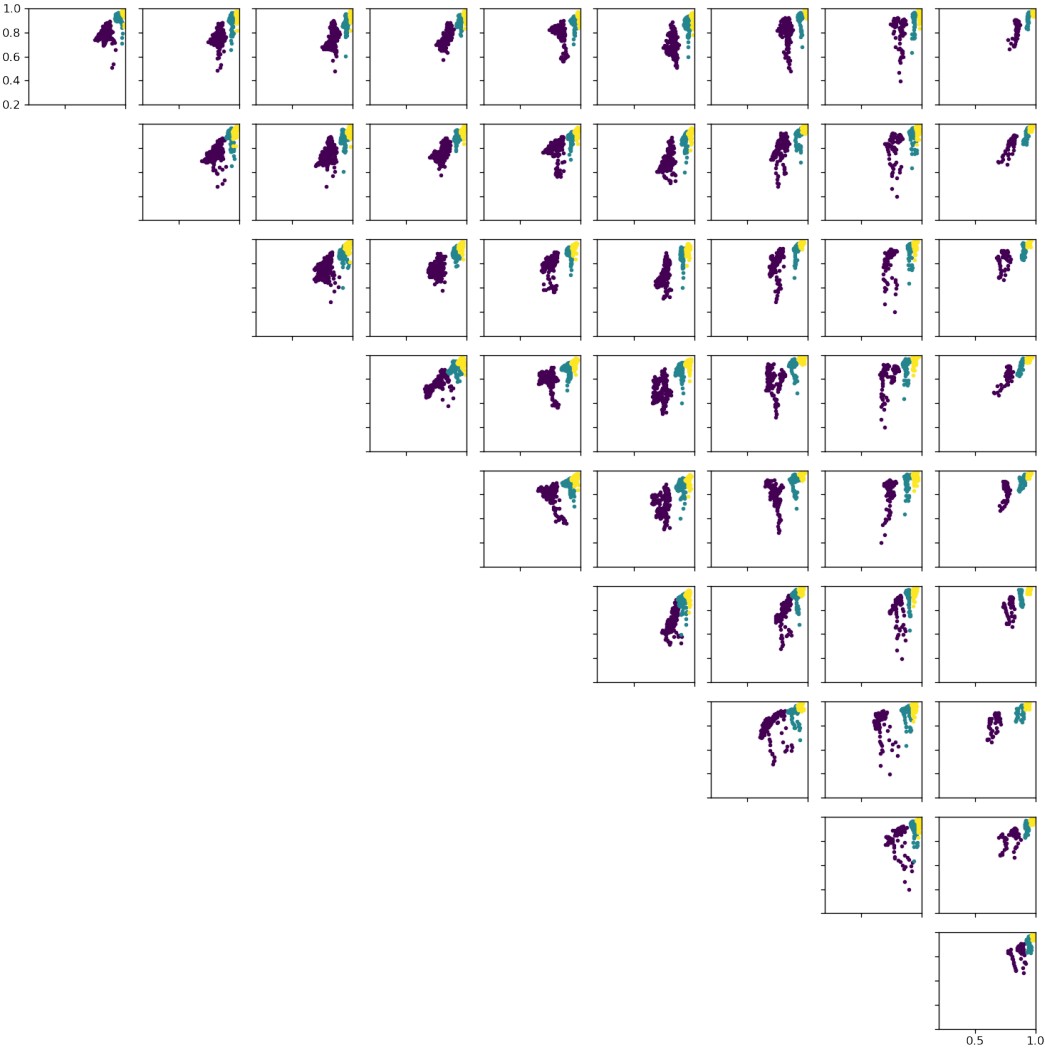

Figure 16: **Forgetting frontiers for 10-task Split CIFAR100, Vision Transformer**. Colors are the same as in Figure 1 of the main text. All images in each row share the same task for the x-axis, i.e. the x-axis of the top row represents task A, x-axis of the second-from-the-top row represents task B, etc. All images in each column share the same task for y-axis, i.e. the y-axis of the leftmost column represents task B, the y-axis of the second-to-leftmost column represents task C, etc.

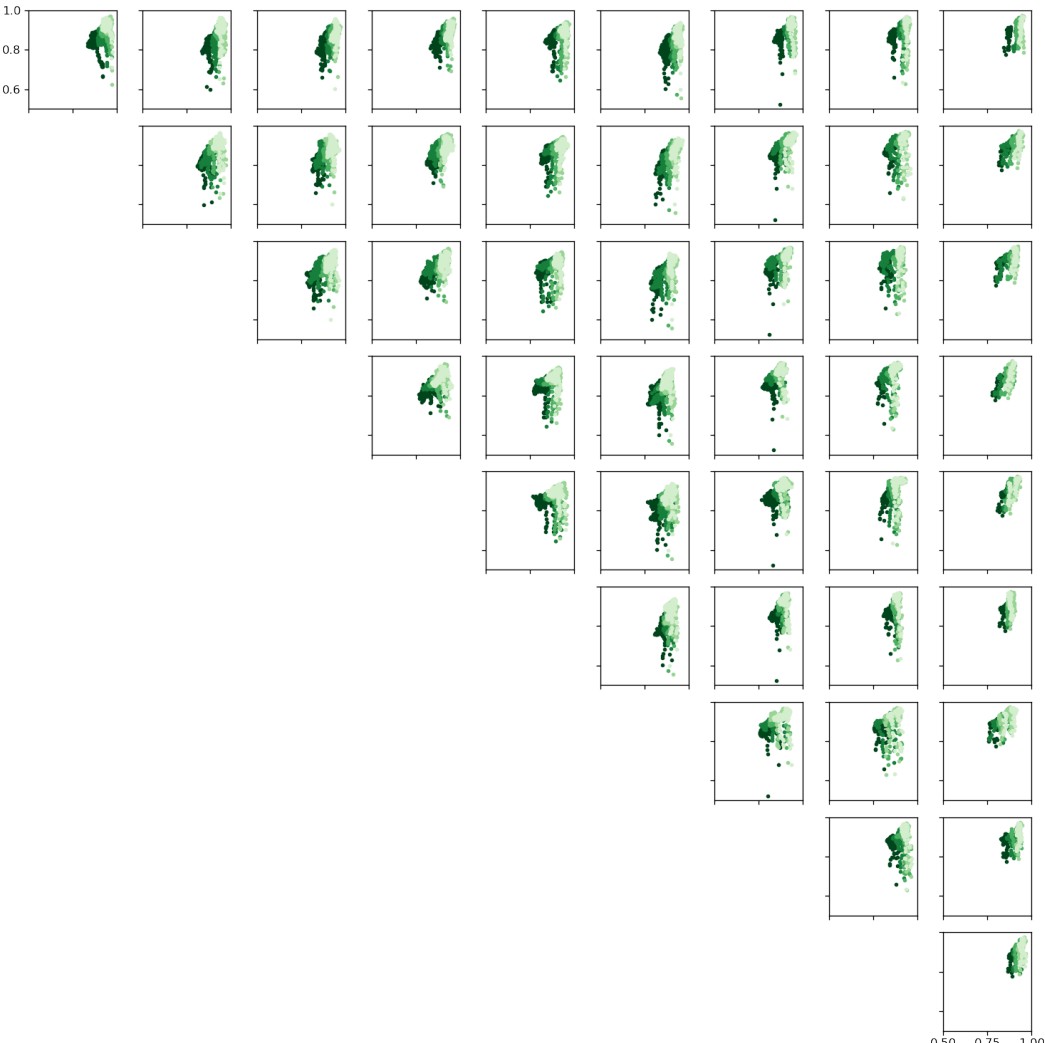

Figure 17: **Forgetting frontiers for 10-task Split CIFAR100 versus data scale**. These frontiers are plotted for a ViT-S_16 model. Colors are the same as in Figure 8(left) of the main text. All images in each row share the same task for the x-axis, i.e. the x-axis of the top row represents task A, x-axis of the second-from-the-top row represents task B, etc. All images in each column share the same task for y-axis, i.e. the y-axis of the leftmost column represents task B, the y-axis of the second-to-leftmost column represents task C, etc.

### B.4 FORGETTING IN DOWNSTREAM TASK SEQUENCES ON DATASETS OTHER THAN CIFAR10 AND CIFAR100

In this section, we study the continual-learning performance of ImageNet21k-models on downstream task sequences beyond CIFAR10 and CIFAR100. Specifically, we use the following datasets:

- The **Oxford-IIIT pet** dataset (Parkhi et al., 2012), a 37-category dataset of images of pets, labeled by breed, with roughly 200 images per class. In our experiments, we create a two-task split of this dataset, with the first 19 breeds (alphabetically by English label) comprising task A and the second 18 breeds comprising task B.

- The **Oxford Flowers 102** dataset (Nilsback & Zisserman, 2008), a 102-category dataset consisting of flowers common in the the UK, labeled by species. In our experiments, we create a two-task split of this dataset, with the first 51 flower species (alphabetically by English species name) in task A and the second 51 flower species in task B.

- The **Street View House Numbers (SVHN)** dataset (Netzer et al., 2011), a 10-category dataset where images consist of single-digit numbers from house numbers. We use the the two-task split in which task A consists of digits 0,1,2,3,4 and task B consists of digits 5,6,7,8,9.

- The **Caltech-UCSD Birds 200 (CUB-200)** dataset (Welinder et al., 2010), a 200-category dataset consisting of photos of birds, labeled by species. As above, we create a two-task split by alphabetically sorting English species names and taking task A to consist of the first 100 categories, and task B to consist of the second 100 categories.

- The **Cars196** dataset, a 196-category dataset consisting of natural images of cars (Krause et al., 2013), with class labels specific models (including years) of cars, e.g. 2012 Tesla Model S or 2012 BMW M3 coupe. We split this into two tasks by using the standard label numbers in TensorFlow datasets, with labels 0-97 in task A and 98-195 in task B.

- The **Domainnet/Clipart** dataset, a subset of **Domainnet** (Peng et al., 2019), consists of clipart drawings (unlike all of the above datasets, which consist of natural images) of 345 different categories including tennis racquets, diamonds, and umbrellas. We create a two-task split alphabetically, with 173 categories in task A and 172 categories in task B.

We use the same basic setup as in the split-CIFAR10 task sequence, in which the first task is trained with a 5-step warmup followed by a cosine decay, while the second task is trained with a constant learning rate. The specific learning rates we use for each of these datasets are given in table 4.

Forgetting frontiers and best-average-error scaling plots are shown in Figure 2. From these plots, we can see that the improvement in forgetting with scale is not unique to the CIFAR10 and CIFAR100 task sequences we used in the main text, but occurs for all of these datasets as well. Scaling plots show that this improvement in average-task performance is a power-law function of the model parameters, but with exponents which vary across datasets.

The broader implication of these plots is that the improvement in forgetting performance with scale is not limited to downstream datasets which are essentially in-distribution of the pretraining dataset (e.g., CIFAR10 and CIFAR100 are very similar to images found in ImageNet21k). As a proxy to quantify how 'in-distribution' the downstream datasets are, in Table 5, we show performances for both a ResNet and ViT model when we train just a linear classifier on top of weights frozen after the ImageNet21k pretraining. We take the closeness of this head-only performance to the full finetuned performance as a proxy for the dataset being in-distribution to ImageNet21k; using this proxy, the Oxford-IIIT Pet dataset, Oxford Flowers dataset, CIFAR10, CIFAR100, and CUB Birds 200 dataset are roughly in-distribution, while Domainnet/Clipart, Cars196, and SVHN are not. But, as we emphasized above, even for these 'out-of-distribution' datasets, forgetting is largely mitigated by scale in the pretrained models.

For completeness, we show head-only and full-finetuning performances for *all* ResNet and ViT models on the first task (task A) of our split-CIFAR10 sequence in Table 6.

| | Vision Transformers | ResNets |
|---|---|---|
| **Oxford-IIIT pet** | | |
| Task A max learning rate | 0.05 | 0.01 |
| Task B learning rates | 0.01, 0.007, 0.003 | 0.003, 0.001, 0.0003 |
| Finetuning steps | 500 | 500 |
| **Oxford Flowers 102** | | |
| Task A max learning rate | 0.03 | 0.003 |
| Task B learning rates | 0.003, 0.001, 0.0003 | 0.0007, 0.0003, 0.0001 |
| Finetuning steps | 500 | 200 |
| **SVHN** | | |
| Task A max learning rate | 0.1 | 0.03 |
| Task B learning rates | 0.06, 0.04, 0.01, 0.007, 0.003 | 0.01, 0.007, 0.003, 0.0007, 0.0003 |
| Finetuning steps | 500 | 500 |
| **Caltech-UCSD Birds 200** | | |
| Task A max learning rate | 0.05 | 0.1 |
| Task B learning rates | 0.016, 0.008, 0.004, 0.002, 0.001 | 0.03, 0.01, 0.007, 0.003, 0.0007 |
| Finetuning steps | 250 | 500 |
| **Domainnet/Clipart** | | |
| Task A max learning rate | 0.13 | 0.1 |
| Task B learning rates | 0.06, 0.04, 0.01, 0.007, 0.003 | 0.03, 0.01, 0.007, 0.003, 0.0007 |
| Finetuning steps | 500 | 500 |
| **Cars 196** | | |
| Task A max learning rate | 0.1 | 0.08 |
| Task B learning rates | 0.03, 0.01, 0.003, 0.001 | 0.03, 0.01, 0.003, 0.001, 0.0003 |
| Finetuning steps | 1000 | 1000 |

Table 4: **Finetuning settings** used for the two-task experiments on datasets described in section B.4.

| | **Oxford-IIIT pet** | | **Flowers** | | **SVHN** | |
|---|---|---|---|---|---|---|
| | Head-only | Full | Head-only | Full | Head-only | Full |
| ResNet101 | 0.9018 | 0.9202 | 0.97168 | 0.95752 | 0.3822 | 0.8992 |
| ViT-S_16 | 0.8923 | 0.9099 | 0.9792 | 0.9842 | 0.3116 | 0.9229 |
| | **CUB Birds 200** | | **Clipart** | | **Cars196** | |
| | Head-only | Full | Head-only | Full | Head-only | Full |
| ResNet101 | 0.8702 | 0.8652 | 0.3977 | 0.7538 | 0.6212 | 0.7112 |
| ViT-S_16 | 0.8047 | 0.8519 | 0.31 | 0.8056 | 0.4759 | 0.7237 |
| | **CIFAR10** | | **CIFAR100** | | | |
| | Head-only | Full | Head-only | Full | | |
| ResNet101 | 0.8701 | 0.9674 | 0.5890 | 0.6924 | | |
| ViT-S_16 | 0.7813 | 0.9592 | 0.5058 | 0.7977 | | |

Table 5: **Comparison of head-only finetuning and full finetuning on downstream datasets.** Using a pretrained (supervised on ImageNet21k) ResNet101 and ViT-S_16, we train either (a) a linear layer on top of frozen representations, or (b) the full model, on downstream datasets. We take this as a proxy for the similarity between the downstream dataset and ImageNet21k; the closer the head-only performance is to the fully-trained performance, the more similarity we expect.

| | Head Only | Full finetuning | | Head Only | Full Finetuning |
|---|---|---|---|---|---|
| ViT-xS_16 | 0.7676 | 0.9527 | ResNet26 | 0.8767 | 0.9474 |
| ViT-S_16 | 0.7906 | 0.9857 | ResNet50 | 0.9030 | 0.9805 |
| ViT-B_16 | 0.9490 | 0.9937 | ResNet101 | 0.9371 | 0.9896 |
| | | | ResNet152 | 0.9362 | 0.9913 |
| | | | ResNet200 | 0.9557 | 0.9933 |

Table 6: **Comparison of head-only finetuning and full finetuning on Task A of Split CIFAR10.** Accuracies achieved by training a linear classifier on top of frozen pretrained representations on the task A split of CIFAR10 (used in the main text), freezing the features of the pretrained models after training on ImageNet21k.

B.5 FORGETTING ON THE PRE-TRAINING TASK DURING FINETUNING

Given the strong performance of large-scale pretrained models on continual learning tasks we have seen in the main text, it is natural to wonder to what extent performance drops on the original pre-training tasks throughout finetuning. One intuition might suggest that because the pretraining dataset is so large, performance drops on the pretraining task will be minimal throughout finetuning. However, in our experiments, we found that this was not the case. In Table 7, we show the performance drops on the pretraining tasks for all models, in two downstream tasks: CIFAR10 and SVHN. In most cases, we find signficant performance drops, though these are much more pronounced when finetuning on the SVHN dataset than when finetuning on CIFAR10.

| | Just pretrained | Pretrained → Finetuned on CIFAR10 | Pretrained → finetuned on SVHN |
|---|---|---|---|
| ResNet26 | 0.3787 | 0.1233 | 0.0001 |
| ResNet50 | 0.4077 | 0.1433 | 0.0005 |
| ResNet101 | 0.4317 | 0.1680 | 0.0015 |
| ResNet152 | 0.4450 | 0.1940 | 0.0021 |
| ResNet200 | 0.4459 | 0.2080 | 0.0027 |
| | | | |
| ViT-xS_16 | 0.2717 | 0.1859 | 0.0085 |
| ViT-S_16 | 0.3699 | 0.3222 | 0.0776 |
| ViT-B_16 | 0.4765 | 0.4300 | 0.3022 |

Table 7: **Forgetting on the pretraining task after finetuning**. Top-1 accuracy on ImageNet21k after pretraining, and after downstream finetuning. Unlike pretrained models performing continual learning on downstream tasks, it appears that there are significant performance drops on the original ImageNet21k task.

## C SUPPORTING RESULTS

In the following subsections, we present results which are directly supporting experiments or figures in the main text, but not new experiments.

### C.1 COMPARING PRETRAINED AND TRAINED-FROM-SCRATCH RESNETS

In the main text, Figure 6, we showed a comparison between the forgetting frontiers (on the split CIFAR10 task sequence) of trained-from-scratch ResNet models and pretrained ResNet models, where we handicapped the pretrained ResNet models (by training them for fewer steps) so that they matched task-A performance of the trained-from-scratch ResNets. This was done to show that even when task-A performance is the same, pretrained ResNets forget far less than their trained-from-scratch counterparts. For space, in the main text we only showed three ResNet models: ResNet26, ResNet101, and ResNet200. Here, we show the forgetting frontiers for all ResNet models (Figure 18), and we also show (Figure 19) the data plotted slightly differently—with all model sizes plotted on the same graph—to make it apparent that while the forgetting frontiers of the pretrained ResNet models clearly improve with model size, the forgetting frontiers of trained-from-scratch ResNet models do not. Figure 19 also includes a scaling plot of the best average error vs. model parameters, supporting the conclusion that in trained-from-scratch ResNets, scaling does not appear to benefit the continual learning performance.

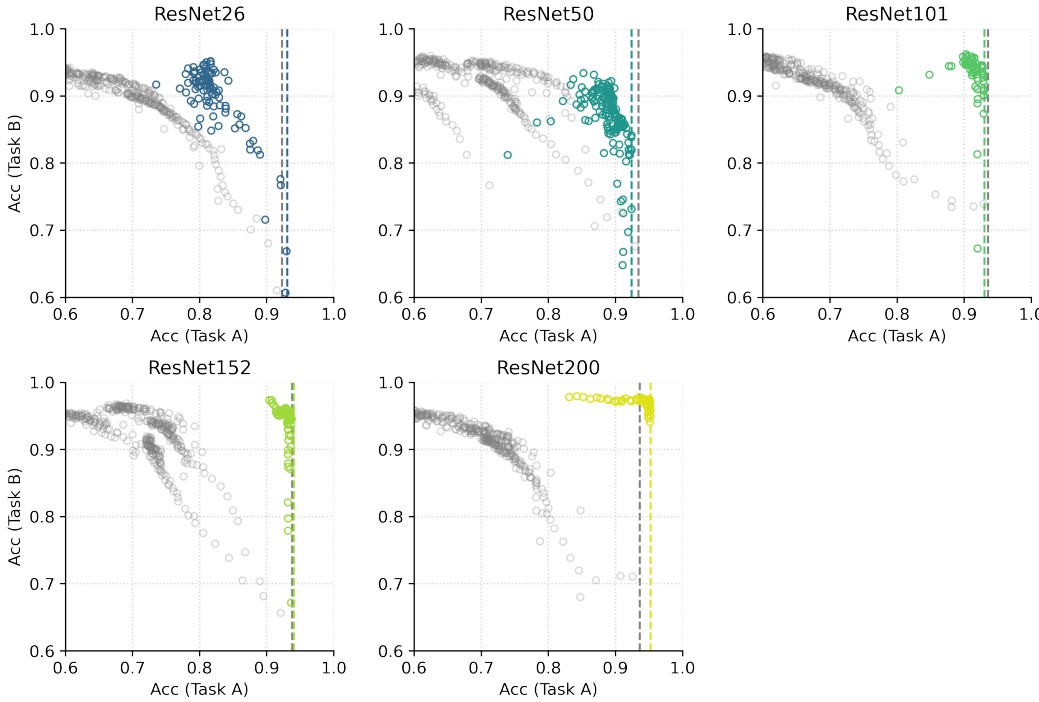

Figure 18: **Forgetting frontier for ResNets trained from scratch, vs. pretrained ResNets (on ImageNet21k)**. The pretrained ResNets were handicapped during fine-tuning, in order to match the Task A performance of the from-scratch model. This figure is identical to Figure 6 in the main text, with the addition of the ResNet50 and ResNet152 models.

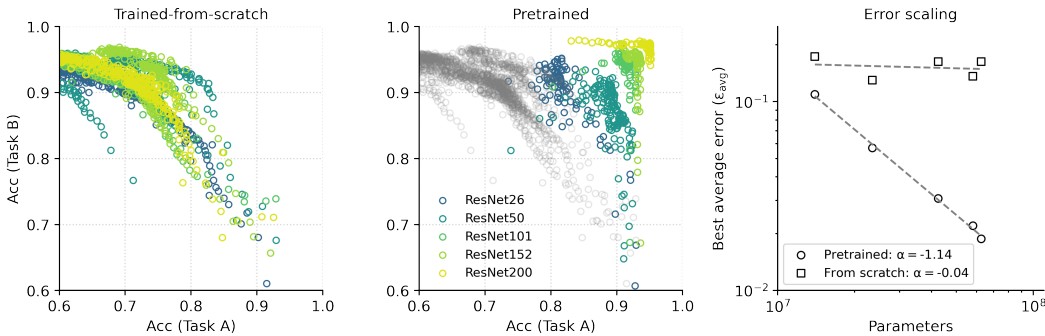

Figure 19: **Forgetting frontier for ResNets trained from scratch, vs. pretrained ResNets (on ImageNet21k)**. The pretrained ResNets were handicapped during fine-tuning, in order to match the Task A performance of the from-scratch model. The data plotted here is the same as that plotted in Figure 6 in the main text; here, the intention is to make clear that the performance of the trained-from-scratch models do not exhibit any apparent benefit with scale, while the pretrained models do (right) Scaling of the best average (task A / task B) accuracy shows that only in pretrained models does the performance improve with scale.

## C.2 LEARNING CURVES FOR SPLIT-CIFAR10 VISION EXPERIMENTS

In Figure 1 of the main text, we plot forgetting frontiers of pretrained Vision Transformers and ResNets on the split-CIFAR10 task sequence. Corresponding to these frontiers are the learning curves (accuracies vs. step) shown in Figures 21 and 20.

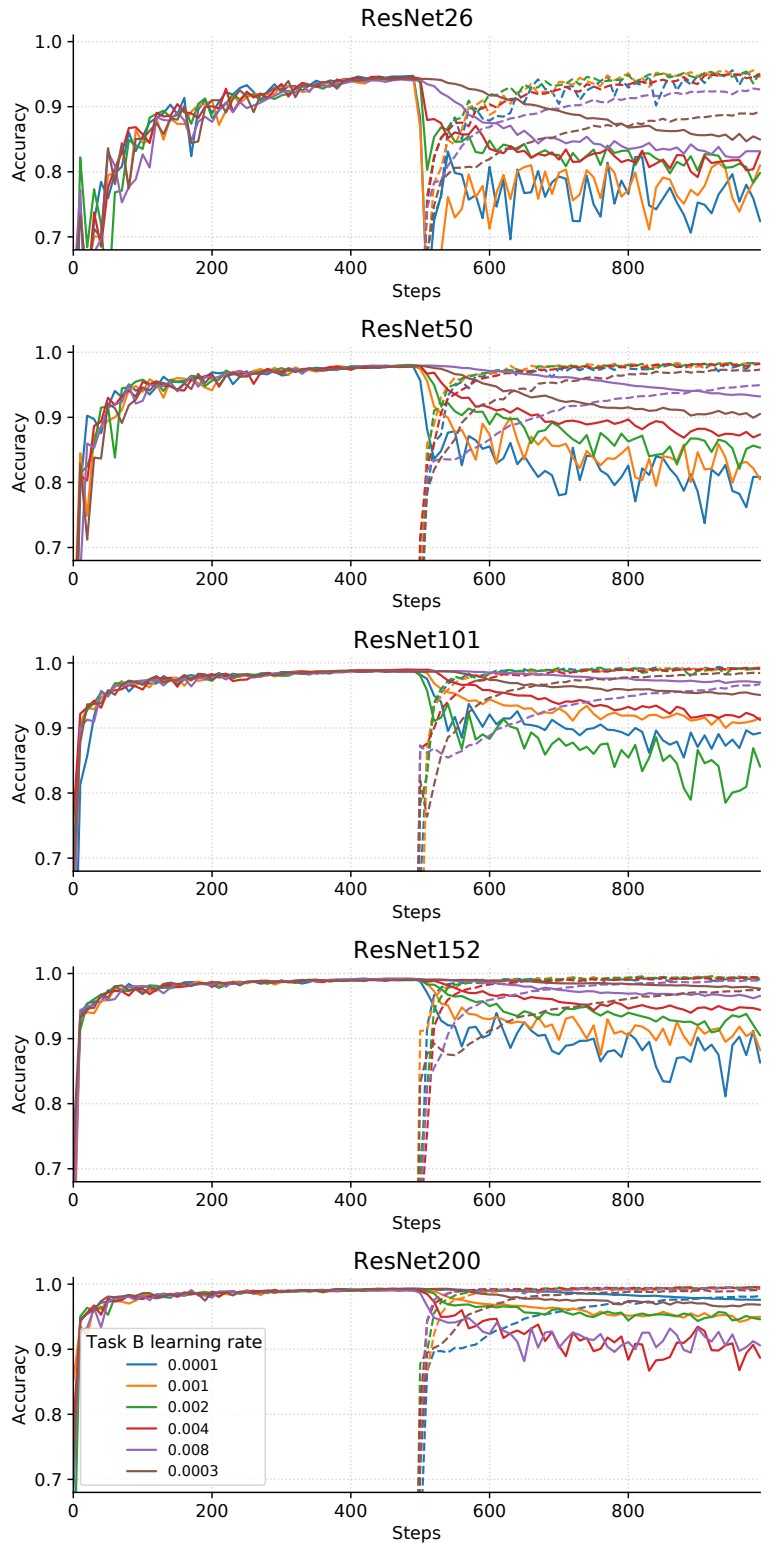

Figure 20: **ResNet Learning curves on Split CIFAR10 task**. Learning curves for the ResNet models of Figure 1. The solid/dashed lines are Task A/B accuracy.

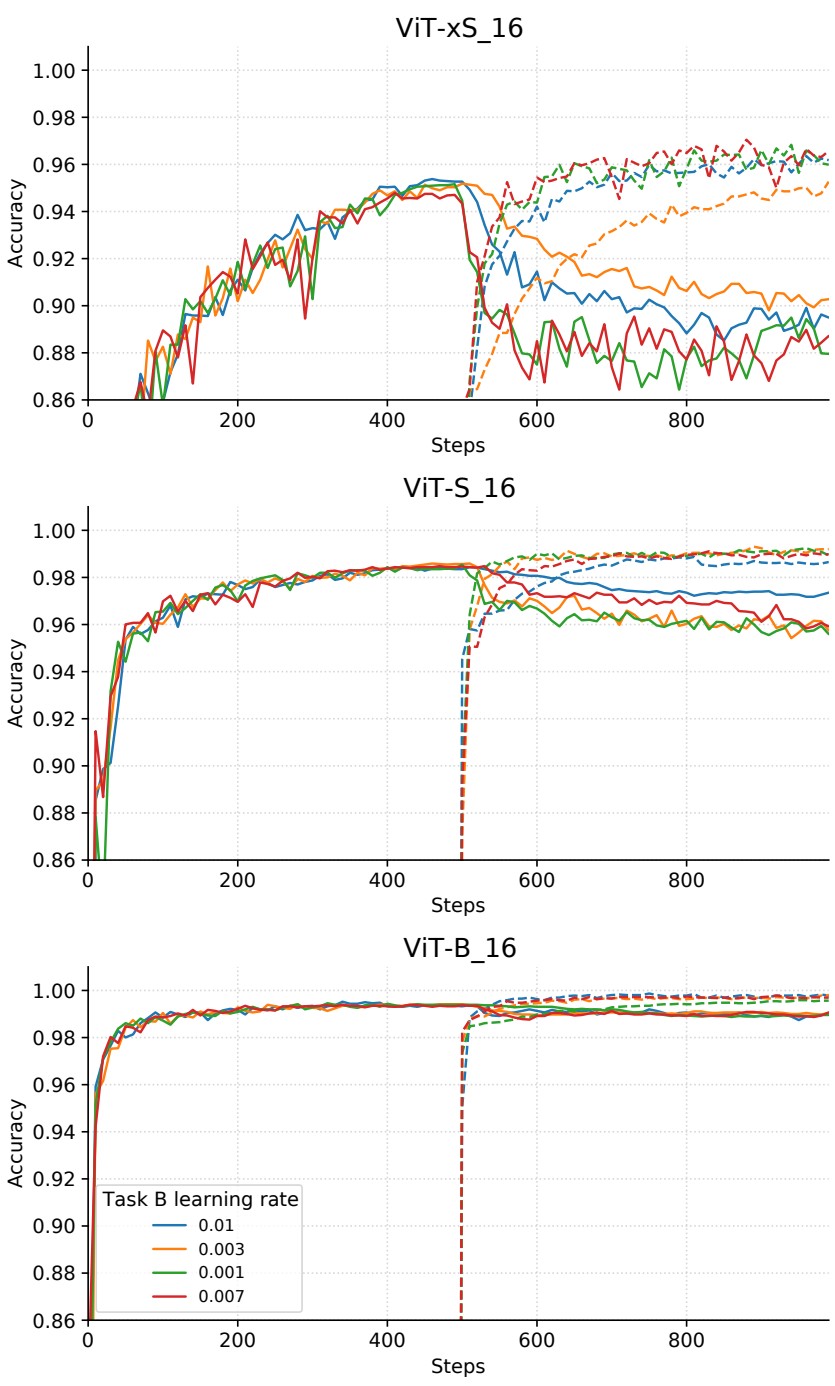

Figure 21: **ViT Learning curves on Split CIFAR10 task**. Learning curves for the ViT models of Figure 1. The solid/dashed lines are Task A/B accuracy.

## C.3 LEARNING CURVES FOR THE LANGUAGE EXPERIMENT

For completeness, Figure 22 displays the token accuracy on task A as a function of the number of steps, corresponding to the frontiers shown in Figure 11 of the main text.

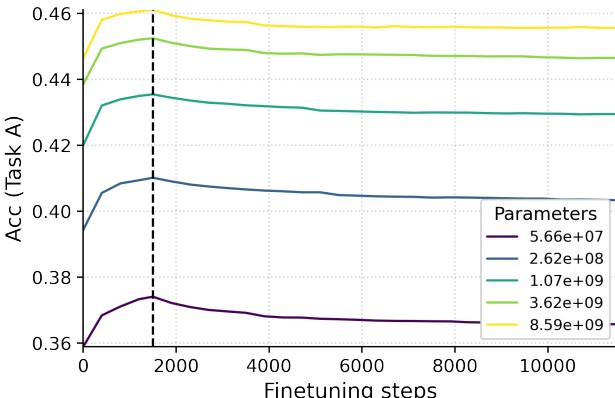

Figure 22: **Learning curves in sequential language-modeling**. Token accuracy as a function of steps for the different models. The dashed line shows the point at which training switches from Task A to Task B.

## C.4 FORGETTING FRONTIERS COLORED BY LEARNING RATE

In the forgetting frontiers of Figure 1 of the main text, the frontiers are colored by model size; each model, however, was trained with several different learning rates during the second task. In Figures 23 and 24, we plot the same frontiers, coloring them by learning rate on the left, and by proper time (learning rate multiplied by step number) on the right.

We also plot a scaling curve in Figure 25 of the average accuracy at a fixed learning rate and step ($\eta = 0.0001$ and $t = 200$), to show that the power-law scaling holds even if one looks at a fixed hyperparameter choice.

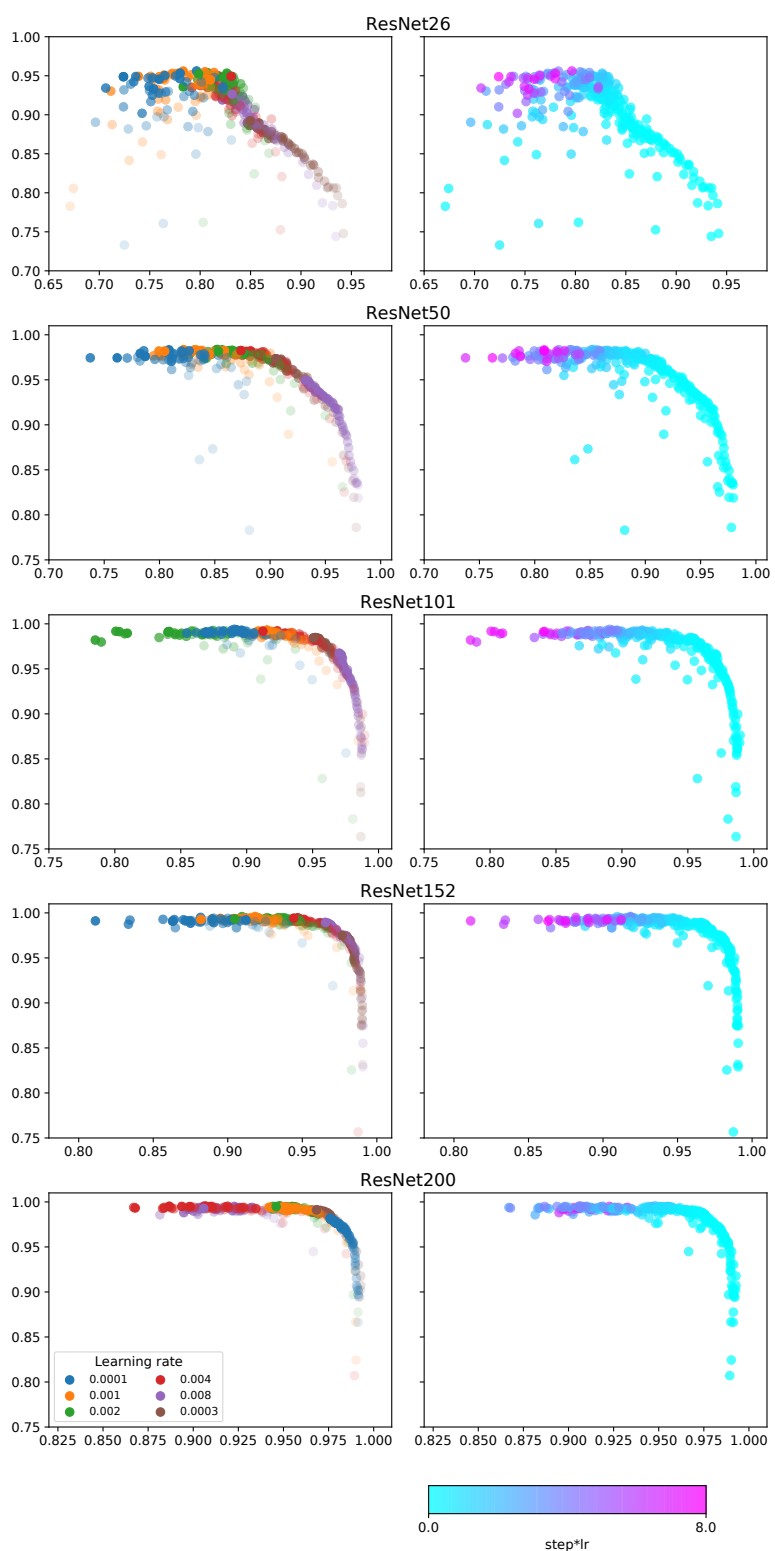

Figure 23: **Step and learning rate along Split CIFAR10 forgetting frontier for ResNets**. We plot the forgetting frontier of Figure 1. The left column is colored by learning rate with transparency set by step (darker) is later during task 2 training. The right column is colored by step times learning rate.

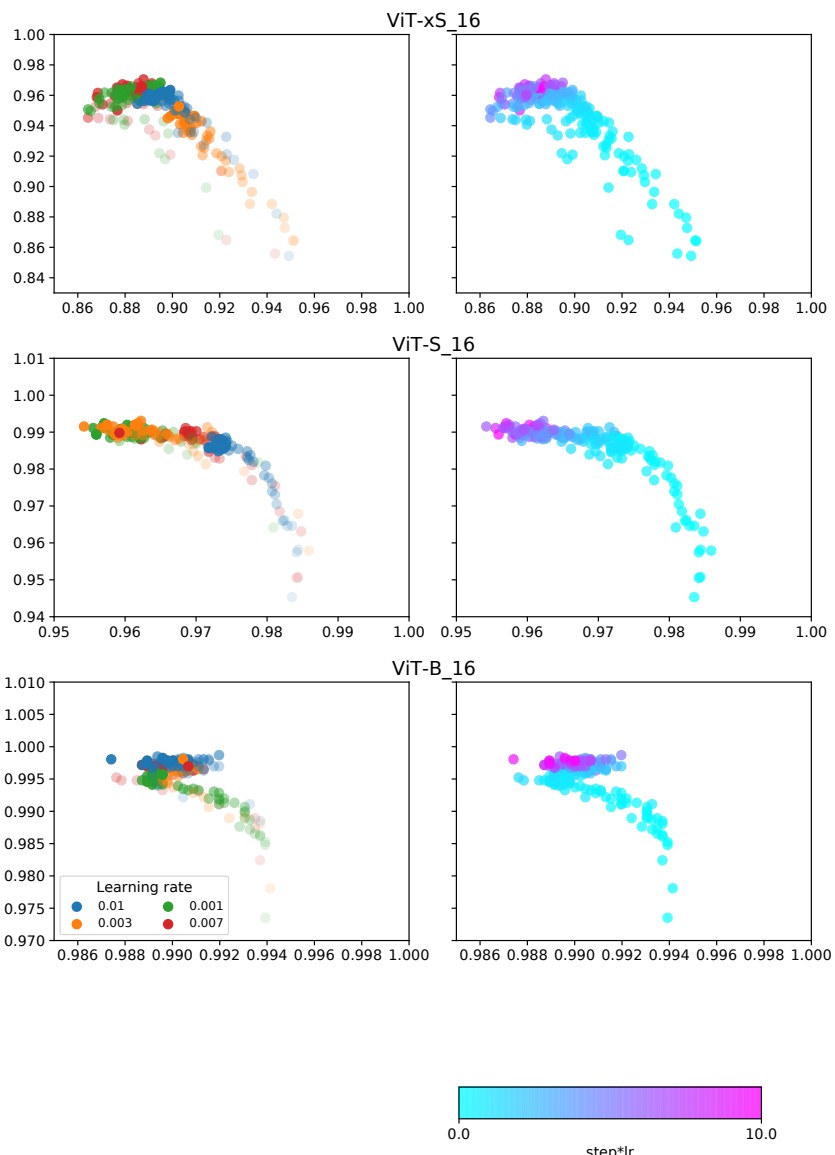

Figure 24: **Step and learning rate along Split CIFAR10 forgetting frontier for ViT models**. We plot the forgetting frontier of Figure 1. The left column is colored by learning rate with transparency set by step (darker) is later during task 2 training. The right column is colored by step times learning rate.

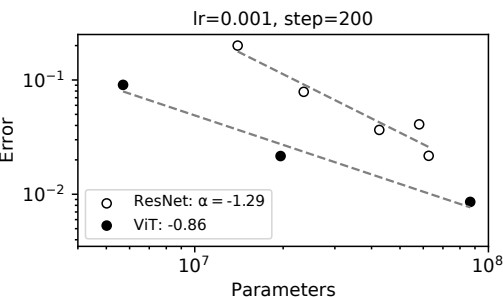

Figure 25: **Average error scaling at fixed learning rate and step**. We plot the average Task A Task B error for the setup in Figure 1 for a fixed learning rate of 0.001 and 200 Task B training steps for both ViTs and ResNets.

## C.5 REPRESENTATION OVERLAP MATRICES FOR ALL RESNET AND VIT MODELS

In the main text, we described the computation of the classwise representation overlap matrices for our models at the end of Task A training, showing an example of such a matrix for both a pretrained and a trained-from-scratch ResNet101 model. In Figure 27, we show representation overlap matrices for all of the ResNet models, which bears out our statement in the main text that pretrained ResNet representations were much closer to orthogonal than their trained-from-scratch counterparts. In Figure 26, we show representation overlap matrices for the (pretrained) Vision Transformer models. As with the pretrained ResNet models, the cross-class representation overlaps become closer to orthogonal as we increase the model size.

In these figures, we also plot the representation overlaps for pre-trained models *before* being trained on task A (which was not plotted in the main text). From these plots one can see that while pre-training on ImageNet21k introduces some level of orthogonality in the model representations of CIFAR10 images, this orthogonality becomes much more pronounced after training the model on task A of the CIFAR10 split.

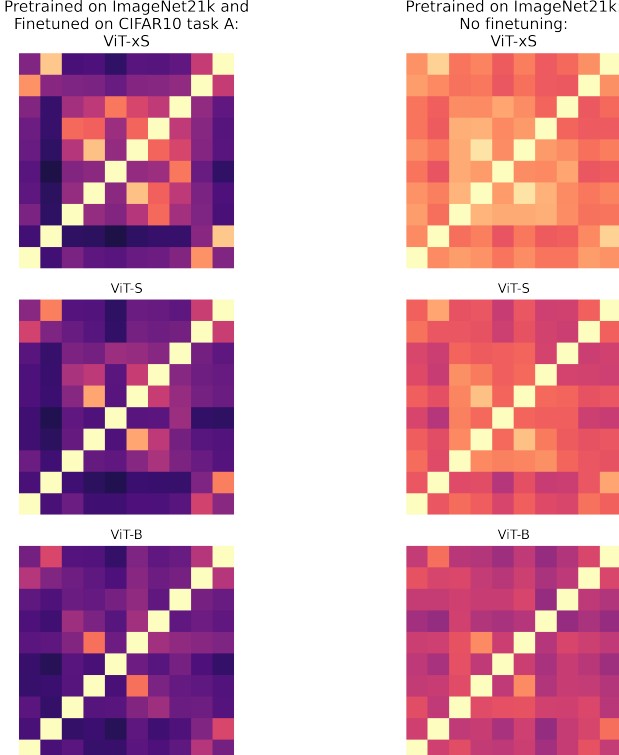

Figure 26: **Classwise representation matrices for Vision Transformer models** taken (left) after the end of training on the first task (Task A) of a two-task split CIFAR-10 sequence, and (right) after pretraining on ImageNet21k. Classes are ordered alphabetically, left to right and down to up, i.e. the order is *airplane*, *automobile*, *bird*, *cat*, *deer*, *dog*, *frog*, *horse*, *ship*, *truck*.

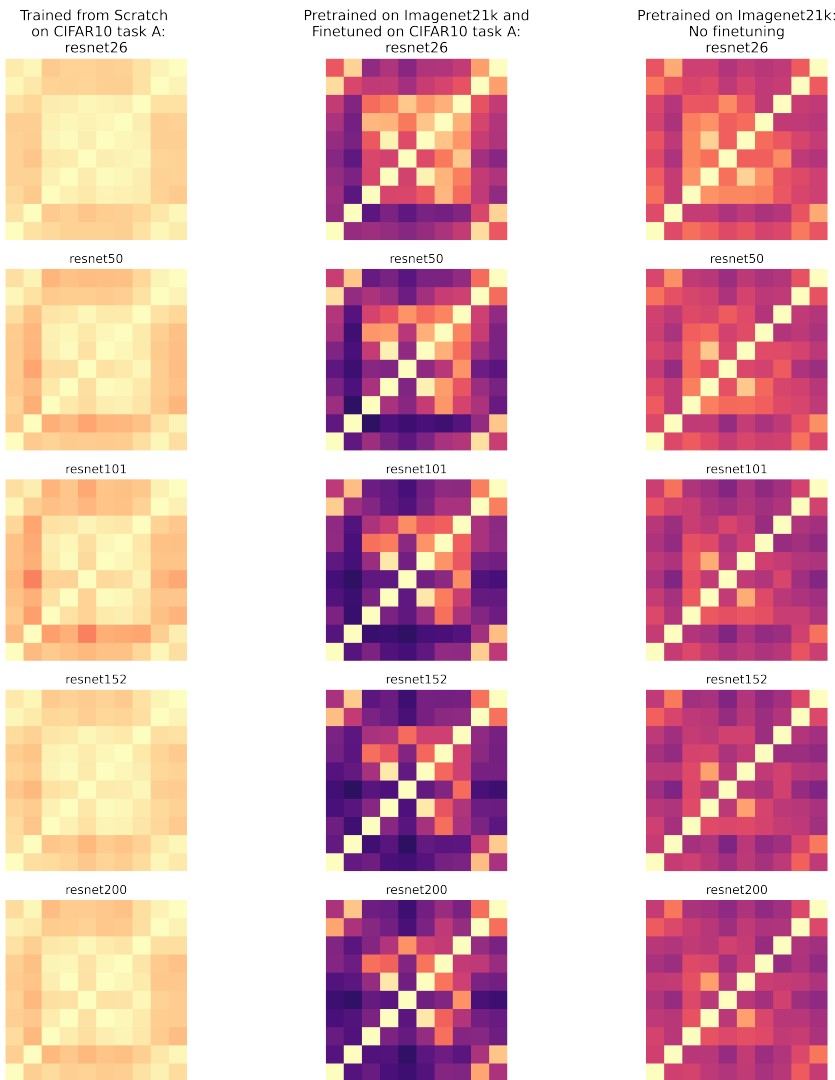

Figure 27: **Classwise representation matrices for ResNet models.** (Left) Models are trained from random initialization on the first task (task A) of a two-task split CIFAR-10 sequence. (Middle) Models are pretrained on ImageNet21k and then finetuned on task A of the two-task split CIFAR-10 sequence. (Right) Models are pretrained on ImageNet21k, but no finetuning is done. Classes are ordered alphabetically, left to right and down to up, i.e. the order is *airplane*, *automobile*, *bird*, *cat*, *deer*, *dog*, *frog*, *horse*, *ship*, *truck*.

