# OpenReview forum: "Effect of scale on catastrophic forgetting in neural networks"
_ICLR.cc/2022/Conference — ICLR 2022 Poster_

### Official Review · Reviewer_Dwjt · 2021-11-01

**Correctness:** 3
**Technical Novelty And Significance:** 2
**Empirical Novelty And Significance:** 3
**Recommendation:** 8
**Confidence:** 4

**Main Review:**

**Strengths:** I found the paper to be well motivated. Recent works on scaling laws for transfer learning primarily focus on understanding the effects of pretrained representations when learning a single new task, but this paper extends the theme of model scaling to understand its benefits when learning multiple tasks. The experiments are extensive: authors evaluate several architectures, different pretraining paradigms (supervised and self-supervised), the impact of pretraining dataset size, amount of fine-tuning, and also show comparisons with training from scratch. I liked the representational similarity figures, which indicated representations across classes become more orthogonal with model scaling, which is likely to prevent catastrophic interference when learning new tasks.

**Weaknesses:** My main apprehensions focus on the paper’s experimental setup and some missing details in the figures. I would be happy to raise my score if these two concerns are addressed.
1. **Overlap of data distributions in pretraining vs. new tasks:** The authors pretrain several models on ImageNet-21K and evaluate on split CIFAR-10/-100 tasks (primarily). However, the two datasets have a significant class overlap and I am uncertain if one can regard training on a subset of CIFAR classes as a new task. If the pretraining data and subsequent tasks’ data have high overlap, we are unlikely to see catastrophic interference when new tasks are learned. This is also highlighted by Figure 8 of the paper, which shows that the pre-trained models have orthogonal representations across CIFAR-10 classes even *without* training the model on new tasks (please correct me if I misunderstood and the models have been fine-tuned on the new tasks). While I understand CIFAR datasets are standardly used benchmarks in the continual learning literature, past work in the field has either used randomly initialized networks or models pretrained on datasets that do not have any overlap with the downstream tasks’ data. Papers that have used ImageNet pretraining [A, B] in the past generally used datasets such as CUB Birds, Oxford Flowers, Places 365, Stanford Cars, etc. for downstream evaluation of catastrophic forgetting. *I believe it is important to include experiments on such non-/minimally- overlapping datasets to ensure subsequent tasks can be considered “new tasks”, thus ensuring better validity of the conclusions drawn from this paper.* Even for the current experiments, I think it is important that the overlap of pretraining data vs. CIFAR data is highlighted.

[A] Memory Aware Synapses: Learning what (not) to forget (https://arxiv.org/abs/1711.09601)

[B] PackNet: Adding Multiple Tasks to a Single Network by Iterative Pruning (https://openaccess.thecvf.com/content_cvpr_2018/papers/Mallya_PackNet_Adding_Multiple_CVPR_2018_paper.pdf)

2. **Training time, learning rates, and figures:** Most figures currently show accuracies of two sequentially learned tasks (e.g., 5-class CIFAR-10 splits, called Tasks A and B). Each datapoint in these figures represents a model trained with either a different learning rate or fine-tuned for a different amount of time--however, all points corresponding to the same model architecture have the same color and marker width. This makes it hard to ascertain whether fine-tuning longer on task A reduces the model’s ability for further transfer to task B, or if there are any improvements/degradation in backward or forward transfer between the two tasks as model size is scaled. Similar comments hold in regards to the effects of different learning rate scales: a large learning rate can often diminish the benefits of pretraining and it is valuable to highlight which points in the figure  correspond to which learning rates. Importantly, I note that I could not find exactly what set of learning rates was used for fine-tuning. *Overall, I think properly delineating the role of these factors is important because the paper’s main message can be misconstrued as saying scaling solves catastrophic forgetting despite how long the model is trained on a set of tasks or what hyperparameters are used.* I note that addressing this comment should primarily involve only replotting the figures by using different shades of a given color, to denote different time steps or learning rates while fine-tuning on Task B.

**Major Questions/Comments**

1. Was the pretraining data pre-processed? ImageNet-21K is heavy tailed and imbalanced, so papers often create a balanced version of the dataset by removing abundant data or classes with too few samples. If pre-processing was performed, this is an important detail that should be explicitly mentioned in the paper. If pre-processing was not performed, I would like to collect the authors’ thoughts on whether the imbalanced nature of this dataset affected their results. For example, would you expect better or similar results with a balanced version? I also highlight that these details are important and should at least be mentioned in the appendix.

2. Change in backbone representation vs. classifier output from final linear layer: My understanding of Figure 8 is that it shows cross class representational similarity after pre-training, before any new tasks are learned. However, when the new tasks are learned, how much does the representation from the backbone change? Is it possible that with effective pre-training, the model representations remain more or less constant and only the last linear layer's output is changing? If this happens, then one possible confounding conclusion is that the benefits of scale are primarily enabling better pre-trained representations. One way of testing this may be to freeze the model and train only a linear layer on top of the model representations. This will be similar to multi-head training, but here the representations will not be allowed to change by preventing fine-tuning.

3. Is self-supervised pre-training performing relatively worse than supervised pre-training? This would be in contrast with self-supervised learning literature’s results, which generally find self-supervised pretraining to be more effective for downstream transfer. Can the authors comment more on this?

**Minor comments**
1. In related work, I think methods that utilize network pruning or model compression techniques to alleviate catastrophic forgetting/interference should be cited (e.g., see [A, B, C] below). Such methods directly utilize the overparameterization of a model and are perhaps most related to the theme studied in this work. Also relevant are works focused on tuning side-networks, where a pretrained representation remains fixed but small side models are used to improve the representation quality for a given dataset.

[A] PackNet: Adding Multiple Tasks to a Single Network by Iterative Pruning (https://openaccess.thecvf.com/content_cvpr_2018/papers/Mallya_PackNet_Adding_Multiple_CVPR_2018_paper.pdf)

[B] Supermasks in superposition (https://arxiv.org/abs/2006.14769)

[C] Side-Tuning: A Baseline for Network Adaptation via Additive Side Networks (https://arxiv.org/abs/1912.13503)

2. On page 6 and 9, the names of recent self-supervised learning methods such as DINO and BYOL are mentioned, but the corresponding papers are not cited.

3. Since the paper’s analysis primarily focuses on pretrained networks and since the authors also comment “pretraining is required to see the benefits of scale”, I think explicitly including the term “pretraining” in the title will be useful.

**Summary Of The Paper:**

Following recent works studying the effect of model scaling on tasks such as transfer learning, this paper asks whether scaling up the size of a pretrained network also helps alleviate or minimize catastrophic forgetting. The authors use split CIFAR-10/-100 datasets (early experiments on NLP tasks are also included) and demonstrate the accuracy lost by a pretrained network when learning a set of tasks sequentially reduces with increase in model size. This result is shown to hold for both ResNets and Vision transformers. The topic is timely and the paper is well written. My main apprehensions relate to the experimental setup--due to the use of only CIFAR datasets for evaluation, which have significant class overlap with ImageNet, it may not be justified to think of learning a subset of CIFAR classes as a new task. Overall, I believe some claims are not completely validated yet, but this can be addressed by using a more appropriate experimental setting.

**Summary Of The Review:**

I found the paper to be interesting and timely, but recommend rejection for now due to a need for better experimental setup and more details in the figures. I would be happy to raise my score if the authors justifiably address these apprehensions.

**Post Rebuttal**: The authors provided detailed response to the raised concerns (though primarily in the appendix) and sufficiently backed up their proposed hypotheses. I am hence increasing my score to recommend acceptance for the paper.

---

> ### Comment · Reviewer_Dwjt · 2021-11-21
> **Thank you for your response; please make sure final paper has experiments from the appendix in the main paper.**
>
> Thank you to the authors for putting significant efforts for addressing my concerns. The newly added experiments are thorough and provide sufficient details to confirm the vital points made in this work. Given this, I am increasing my score, but want to point out an important limitation in the manuscript right now: While the new experiments resolve all raised questions, these experiments are primarily located in the appendix. It will be vital to ensure some of these experiments are moved to the main paper, and, in fact, take the main stage instead of the existing figures in the main paper. For example, Column 2 of Figure 23/24 are really useful presentations of Figure 1 and I think it makes a lot of sense to have some of those sub-figures in Figure 1 itself. I also strongly encourage focusing on non-CIFAR datasets for making your points in the main paper. Using CIFAR datasets for studying the problem of catastrophic forgetting may not be appropriate in continual learning with ImageNet-pretrained model settings (though I accept it turned out to be valid in this situation).

---

### Official Review · Reviewer_r2yB · 2021-11-02

**Correctness:** 3
**Technical Novelty And Significance:** 3
**Empirical Novelty And Significance:** 3
**Recommendation:** 8
**Confidence:** 5

**Main Review:**


**Strengths**:
(1) The paper is well-written and easy to follow.
(2) The main motivation of the paper (i.e., role of scale and pre-training) is significant.

**Weaknesses**:

(1)  I think the major problem with the arguments of the paper is that it mixes the over-parametrization and pre-training together.
For instance, the paper mentions: “catastrophic forgetting is mitigated to a large extent by scale: that is, larger models suffer less from forgetting”, which is not a true argument in my opinion.
The high-performance is due to the “pre-training” part and not the over-parametrization. In fact, [2] shows that over-parametrization by increasing depth is not very helpful for continual learning.
This is also visible in Fig 4. where ResNet 26 and ResNet 101 perform almost equally poorly, which suggests over-parametrization is not very helpful.

(2) If we agree on the fact that the benefit comes mostly from pre-training, then the other problem is that this has also been studied before [1] for both vision and NLP benchmarks.
So I believe the fact that pre-training is helpful is not a new phenomenon in the CL literature.

(3) The final major problem is with the experimental setup.
First, pre-training on the ImageNet-21K benchmark and then continual learning on a similar dataset makes the evaluation very tricky.
If we look at the classes in ImageNet-21K [3], we can see that it shares many classes to CIFAR and ImageNet-1K, and the pre-trained model had already seen data from different distributions before.
So the model already knows the distribution of vehicles, animals, foods, etc., and then the forgetting is not well-defined here.
I believe a correct pre-training setup should be what people are doing in the NLP literature and what authors have done in Section 3.4.
When pre-training, the source task should be different than the fine-tune task. For instance, in self-supervised prompting, the model learns the representations by figuring the masked words and learning a language model.
Then, in that setup, one can argue that the knowledge of the model is related to the language model, which can be used in different downstream tasks with different data.
However, here, the model is pre-trained on the same task (image classification) on a dataset that potentially has the labels of the fine-tune task.
In general, while I have no doubt regarding the benefits of pre-training, I believe the experimental setup should be more carefully designed.


[1] Mehta et al. “An Empirical Investigation of the Role of Pre-training in Lifelong Learning”, ICML Theory of CL Workshop (2021).
[2] Mirzadeh et al. “Wide Neural Networks Forget Less Catastrophically.” ArXiv abs/2110.11526 (2021).
[3] “WordNet names”, [https://storage.googleapis.com/bit_models/imagenet21k_wordnet_lemmas.txt](https://storage.googleapis.com/bit_models/imagenet21k_wordnet_lemmas.txt)

---------
*** **UPDATE AFTER DISCUSSION PERIOD** ***
After reading other reviews, authors' responses, and the updated version of the paper, I believe major concerns have been addressed. Thus, I have increased my initial score.

**Summary Of The Paper:**

The paper studies the impact of large-scale pre-training in continual learning.
Moreover, The paper shows that pre-trained models suffer less from forgetting in vision and NLP tasks with various experiments.


**Summary Of The Review:**

I believe the paper lacks novelty and some claims are not well-supported. Moreover, the experimental design has flaws as mentioned in the main review.

---

### Official Review · Reviewer_ZtJq · 2021-11-03

**Correctness:** 2
**Technical Novelty And Significance:** 2
**Empirical Novelty And Significance:** 3
**Recommendation:** 5
**Confidence:** 4

**Main Review:**

Strengths

-The authors open an interesting and timely discussion about the size of models and pre-training and how they affect catastrophic forgetting and lifelong/continual learning

-The use of the forgetting frontier is an interesting way to visualize two task continual learning settings

-Several model classes are used and the authors provide insights about the importance of pre-training


Weakness/Comments

-The choice of pre-trained model and the set of tasks is odd. CIFAR-10 is an extremely similar task to imagenet and using imagenet pre-trained model features directly should do well. The authors should include the obvious baseline of just training the head on fixed features on their graphs. It is already known that using better pre-trained models particularly for such tasks will improve performance on the down stream task. Similarly Pretraining imagenet with SIMCLR is known to give strong features for object recognition and is more over done on essentially an input data distribution that more or less includes CIFAR-10, despite different training objective.  So the proposed setup for very powerful imagenet models seems like it will mostly be fighiting changing the model weights in the first task.

-The similarity of the pre-trained task and the task sequence is not properly decoupled in this work. Should we conclude that pre-training alone is the key or pre-training on very similar data distributions to the sequence that follows? I would have liked to see at least one experiment with a legitimately diverse input distribution from the pretraining distribution (e.g. different object categories from previously seen or non-natural images).

-Many details of the  forgetting frontier points don't seem to be provided, for example the learning rates used or other optimization hyperparameters don’t even seem to be provided at all in the paper. Furthermore the learning rates and finetuning steps are mixed in the forgetting frontier plots making it challenging to fully interpret the results visualized. Do the larger models typically have smaller learning rates as the points with better taks 1 and taks 2 trade of? It also will make it challenging to reproduce the results. I think this visualizations should more clearly control for either finetuning step or learning rate.

-Standard evaluation protocols for continual learning are not considered (e.g. forgetting or reporting an overall accuracy with a clearly stated hyperparameter search criteria). The use of the forgetting frontier is interesting but its one view and may not give a full picture. We know a larger pre-trained model on imagenet will have better performance on downstream tasks (e.g. Kornblith et al Do Better Imagenet Models Transfer Better?) so we know the first task will generally do better in this setting as the model has better imagenet performance, how do we clearly compare now the forgetting in this case where we know the first task will do better? In the experiments comparing to from scratch performance the authors handicap the models for the task A to start at the same point, can similary idea be used in other experiments somehow?

-The use of pre-training seems to suggest we should evaluate forgetting on the imagenet-21k as well to get more insight into the behavior. Have the authors consider this?

-The observation that the model size doesn't clearly improve forgetting when training from scratch is interesting. What if the first task is big (e.g. 1/4 of the imagenet-21k data), if followed with a 2 and 3 task how is this type of sequence really different from the pre-training formulation here?  A more natural setup seems to be to split imagenet-21k into a series of tasks and assume the first task is the “pre-trained” model. This would also end up with a situation where the CL sequence distribution does not clearly follow the pre-trained data distribution by separating them via object categories.


**Summary Of The Paper:**

The authors study using ever larger pretrained models and suggest this can solve the problem of catastrophic forgetting without the use of any particular continual learning method. They support this claim with experiments using pre-trained models on imagenet applied to CIFAR-10 dataset as well as a Natural Language task with a large pre-trained model then applied to IMDB and english Wiki.

**Summary Of The Review:**

Studying the effects of model size and pre-training in continual learning seems pertinent. However, the claims at the moment in the paper seem to be overly general given the evaluation. The paper has experimental setups that make it hard to decouple the effects of various factors including distribution similarity of pre-training to CL sequence data, optimization hyperparemeters, finetuning steps, performance of first task.

---

### Official Review · Reviewer_Tmvz · 2021-11-08

**Correctness:** 3
**Technical Novelty And Significance:** 3
**Empirical Novelty And Significance:** 3
**Recommendation:** 5
**Confidence:** 4

**Main Review:**

**Positives:**
- Overall, I enjoyed the direction of this paper. I think it was asking interesting questions, and the experiments were quite thorough with respect to the scale of the dataset and model
- The results show quite a clear picture in terms of the effects of scale on forgetting in the settings the paper explores.
**Negatives:**
- I would like to see these results extended past the two task setting. It would be interesting to see whether these results would hold for many tasks in sequence. Many datasets used in the current literature often have anywhere from 5-20 tasks, and it's unclear how well the results would hold for those settings.
- The paper shows some results on 10 tasks Split CIFAR-100, but there doesn't seem anything to compare the results to. Only the curves for a (single) pretrained ViT-B model are shown, and it's unclear what the takeaway of that experiment is with respect to the effects of model size, dataset size, or pretraining vs random initialization
**Questions:**
- Is the data used in pretraining controlled for? Specifically, there is quite an overlap in classes between the ImageNet dataset and the CIFAR datasets used. Is the effect of that accounted for?
**Minor:**
- The explanation of the trace overlap metric can be improved.

**Summary Of The Paper:**

The paper explores the effect of scale on catstrophic. They conduct several different experiments (for both ResNets and Vision Transformers) showing:
- the size of the model for pretrained models reduces forgetting
- (Longer) pretraining reduces forgetting
- More pretraining training data size reduces forgetting
- A metric used to measure overlap between model representations showed that pretraining results in more orthoganal representations between classes.
- The length of finetuning isn't as impactful on pretrained models as randomly initialized models
They perform the majority of their experiments on a two task, task incremental Split CIFAR-10 dataset. Some experiments were done on 2-task class incremental and 10-task task incremental Split CIFAR-100. They also conduct a brief set of experiments showing that scaling model size  reduces forgetting for language as well.

**Summary Of The Review:**

This paper does a lot of good things and is fairly comprehensive in terms of trying different model sizes and dataset sizes. My biggest problem with this paper is that it (mostly) only explores the effects of scale on a limited 2 task setting. These results may not hold for longer task sequences that are often used in continual learning, and thus I recommend a borderline reject.

---

### Decision · Program_Chairs · 2022-01-20

**Decision:**

Accept (Poster)

**Comment:**

The authors explore the forgetting behavior of large-scale pre-trained models in continual learning (CL). The authors find that forgetting is mitigated by scale (or models and of pre-training datasets). The authors also make preliminary observations to try to explain why this happens. This manuscript is somewhat in line with recent results around scaling laws and in a sense, this paper extends the study of scaling laws to the CL setup with a focus on catastrophic forgetting, traditionally the main desiderata of CL.

The initial reviewer assessments indicated that this paper was likely below the acceptance threshold of the conference. The main perceived limitations were:
+ CL experiments on sequences of two tasks are limiting (in CL it's common to look at sequences of at least 10 tasks)
+ The effect of the pre-training data on the results was not properly assessed. In particular, it is likely that the pre-training data and downstream/test data were very similar.
+ Other aspects such as missing hyperparameter values and empirical settings were also raised.

The authors really came through and obtained lots of additional empirical results. In particular, the authors show that their results mostly hold on longer task sequences and even if the downstream task was very different from the pre-training tasks. Further, the authors provided precise answers to all the reviewer comments and also ran a few more studies to answer more specific reviewer questions (including a few to answer some excellent suggestions from reviewer ZtJq).

Overall, this is a good contribution and I imagine one that could have a significant impact in the field and give rise to follow-up work. Congratulations!

In preparing the final version of the manuscript, I would strongly suggest that the authors incorporate all results discussed in their replies in their paper. Further, and as was suggested by reviewer Dwjt, I think it would be very useful to add the study of the longer task sequences and of the different downstream tasks to the main paper and not the appendix as is done currently.